# Bounding the Expected Robustness of Graph Neural Networks Subject to Node Feature Attacks

**Yassine Abbahaddou** *
LIX, Ecole Polytechnique
IP Paris, France

**Sofiane Ennadir** *
EECS, KTH
Stockholm, Sweden

**Johannes F. Lutzeyer**
LIX, Ecole Polytechnique
IP Paris, France

**Michalis Vazirgiannis**
Ecole Polytechnique, IP Paris
& KTH Stockholm, Sweden

**Henrik Boström**
EECS, KTH
Stockholm, Sweden

## Abstract

Graph Neural Networks (GNNs) have demonstrated state-of-the-art performance in various graph representation learning tasks. Recently, studies revealed their vulnerability to adversarial attacks. In this work, we theoretically define the concept of expected robustness in the context of attributed graphs and relate it to the classical definition of adversarial robustness in the graph representation learning literature. Our definition allows us to derive an upper bound of the expected robustness of Graph Convolutional Networks (GCNs) and Graph Isomorphism Networks subject to node feature attacks. Building on these findings, we connect the expected robustness of GNNs to the orthonormality of their weight matrices and consequently propose an attack-independent, more robust variant of the GCN, called the Graph Convolutional Orthonormal Robust Networks (GCORNs). We further introduce a probabilistic method to estimate the expected robustness, which allows us to evaluate the effectiveness of GCORN on several real-world datasets. Experimental experiments showed that GCORN outperforms available defense methods. Our code is publicly available at: https://github.com/Sennadir/GCORN.

## 1 Introduction

Graph-structured data is prevalent in a wide range of domains, motivating therefore the development of neural network models that can operate on graphs, known as Graph Neural Networks (GNNs). GNNs have emerged as a powerful tool for learning node and graph representations. Many GNNs are instances of Message Passing Neural Networks (MPNNs) (Gilmer et al., 2017) such as Graph Isomorphism Networks (GIN)(Xu et al., 2019b) and Graph Convolutional Networks (GCN)(Kipf & Welling, 2017). These models have been successfully applied in real-world applications such as molecular design (Kearnes et al., 2016). In parallel to their success, it has been shown, particularly in the field of computer vision, that deep learning architectures can be susceptible to adversarial attacks (Goodfellow et al., 2015). These attacks, which are based on injecting small perturbations into the input, lead to unreliable predictions, limiting therefore the applicability of these models to real-world problems. Similar to other deep learning architectures, GNNs are also vulnerable to adversarial attacks. Recent studies (Dai et al., 2018; Zügner et al., 2018; Günnemann, 2022) have shown that a GNN can be attacked by applying small structural perturbations to the input graphs. These attacks pose a threat to the reliability of GNNs, particularly in safety-critical applications such as finance and healthcare. Consequently, different attacks have been proposed to explore the robustness of GNNs. Concurrently, several studies focus on developing methods to mitigate possible perturbation effects and improve the robustness of MPNNs. The proposed methods include augmenting training data with adversarial examples and retraining the model (Feng et al., 2019), pre-processing methods such as

---

*Equal contribution. Contact: `yassine.abbahaddou@polytechnique.edu` and `ennadir@kth.se`

edge pruning (Zhang & Zitnik, 2020), and more recently robustness certificates (Schuchardt et al., 2021). While the majority of these proposed defenses focus on structural perturbations, only limited advances have been made for feature-based adversarial attacks on graphs. Moreover, despite the large amount of research on the robustness of these methods through empirical exploration, there has been limited progress in understanding the theoretical robustness of GNNs.

In this work, we conduct a theoretical examination of the robustness of MPNNs subject to adversarial attacks. First, we formally define the concept of "Expected Adversarial Robustness" for structure and feature-based perturbations in the context of graphs defined in a metric space. We furthermore establish a formal relation between our introduced expected robustness and the conventional adversarial robustness formulation. Further, by analyzing the input sensitivity of the iterative message-passing schemes, we derive an upper bound on the robustness of these models for both structural and feature-based attacks. Motivated by our theoretical results, we propose a refined learning scheme, called *Graph Convolutional Orthonormal Robust Network (GCORN)*, for the GCN, to improve its robustness to feature-based perturbations while maintaining its expressive power. In addition to our theoretical findings, we empirically evaluate the effectiveness of our GCORN in both node and graph classification on commonly used real-world benchmark datasets and compare GCORN to existing methods to defend against feature-based adversarial attacks. Most empirical evaluations of the effectiveness of defense methods consider worst-case scenarios, hence not taking into account the variability of attacks and their likelihood of occurrence. To overcome this limitation, we propose a novel probabilistic method for evaluating the expected robustness of GNNs, which is based on our introduced robustness definition. The method is model-agnostic and can hence be applied to any architecture to estimate the local robustness. More realistic and comprehensive evaluation of the effectiveness of defense approaches can hence be conducted.

Our main contributions are: **(i)** We define and theoretically analyze the expected robustness of MPNNs, producing an upper-bound on their expected robustness, **(ii)** a novel approach (GCORN) for improving the expected robustness of GCNs to feature-based attacks while maintaining their performance in terms of accuracy, **(iii)** a theoretically well-founded, probabilistic and model-agnostic evaluation method, and **(iv)** an empirical evaluation of our GCORN on benchmark datasets, demonstrating its superior ability compared to existing methods to defend against feature-based attacks.

## 2    RELATED WORK

While most studies on *attacking machine learning models* focus on images (Goodfellow et al., 2015), work on discrete spaces such as graphs has also emerged. Analogous to images, most existing graph-based attack methods frame the task as a search/optimization problem. For instance, the targeted attack Nettack (Zügner et al., 2018) utilized a greedy optimization scheme while Zügner & Günnemann (2019) formulate the problem as a bi-level optimization task and leverage meta-gradients to solve it. Zhan & Pei (2021) expanded this work through a black-box gradient algorithm overcoming several limitations. Furthermore, Dai et al. (2018) proposed to use Reinforcement Learning to solve the search problem and generate adversarial attacks. Node injection attacks (Zou et al., 2021; Tao et al., 2021; Chen et al., 2022; Ju et al., 2023) have also proven effective, with attackers introducing malicious nodes instead of modifying existing nodes or edges, affecting therefore the model's performance. In parallel, the field of *defending against adversarial attacks* on GNNs is still relatively under-explored compared to that of image-based models. The majority of methods are primarily focused on heuristic strategies. Similar to computer vision, robust training (Zügner & Günnemann, 2019) and noise injection (Ennadir et al., 2024) have been used to improve the robustness of GNNs. Additionally, low-rank approximation with graph anomaly detection (Ma et al., 2021) has been used to defend against adversarial attacks. The GNN-Jaccard method (Wu et al., 2019) pre-processes the adjacency matrix to detect potential manipulation of edges. In the same context, GNN-SVD (Entezari et al., 2020) uses a low-rank approximation of the adjacency matrix to filter out noise. Other methods such as edge pruning (Zhang & Zitnik, 2020) and transfer learning (Tang et al., 2020) have also been proposed. Finally, Wang et al. (2020) proposed a low-pass adaptation of the message passing to enhance robustness while providing theoretical guarantees. Although these defense strategies have had some success, for the majority of them, their heuristic nature results in defenses against specific types of attacks without any guarantees on the model's underlying robustness. As a result, these defenses may be susceptible to being circumvented by future new advanced attacks. Consequently, the *investigation of robustness certificates* (Zügner & Günnemann, 2019; Bojchevski & Günnemann,

2019; Gosch et al., 2023) has gained attention by providing attack-independent guarantees on the stability of the model's predictions such as randomized smoothing (Bojchevski et al., 2020).

While the majority of the existing work on defense strategies for GNNs has focused on structural perturbations, there is far less work on addressing feature-based attacks. This represents a significant gap in the literature as feature-based attacks on GNNs can be very effective. Seddik et al. (2022) propose to add a node feature kernel to the message passing to enhance the robustness of GCNs. Additionally, RobustGCN (Zhu et al., 2019) proposes to use Gaussian distributions as the hidden representations, enabling the absorption of the impact of both structural and feature-based attacks. Finally, Liu et al. (2021a) edited the message passing module using adaptive residual connections and feature aggregation, which have been shown experimentally to enhance the model's robustness against abnormal node features. Robustness certificates have also been proposed for node feature-based attacks (Scholten et al., 2022; Bojchevski et al., 2020).

# 3 EXPECTED ADVERSARIAL ROBUSTNESS

In this section we mathematically define the concept of the expected robustness for a graph-based function, such as a GNN. Let us consider three metric spaces with defined norms over the graph space $(\mathcal{G}, \|\cdot\|_{\mathcal{G}})$, the feature space $(\mathcal{X}, \|\cdot\|_{\mathcal{X}})$ and the label space $(\mathcal{Y}, \|\cdot\|_{\mathcal{Y}})$. Let $\mathcal{D}$ be the underlying probability distribution defined on $(\mathcal{G}, \mathcal{X}, \mathcal{Y})$. Given a graph-based function $f : (\mathcal{G}, \mathcal{X}) \to \mathcal{Y}$, and some input $(G, X) \in (\mathcal{G}, \mathcal{X})$ with its corresponding label $y \in \mathcal{Y}$ where $f(G, X) = y$, the goal of an adversarial attack is to produce a perturbed graph $\tilde{G}$ and its corresponding features $\tilde{X}$ which are 'slightly' different from the original input $(G, X)$ such that the predicted class of $(\tilde{G}, \tilde{X})$ is different from the predicted class of $(G, X)$. The adversarial task is contingent on defining a similarity measure between the input graph and the adversarially generated graph. To this end, we introduce a distance over our input metric spaces, which takes both the graph and its corresponding features into account

$$d^{\alpha,\beta}([G, X], [\tilde{G}, \tilde{X}]) = \alpha\|G - \tilde{G}\|_{\mathcal{G}} + \beta\|X - \tilde{X}\|_{\mathcal{X}}.$$

In practice, a graph is represented by its adjacency matrix (or some other graph shift operator) and its feature matrix, we can therefore, without loss of generality, consider the distance

$$d_2^{\alpha,\beta}([G, X], [\tilde{G}, \tilde{X}]) = \min_{P \in \Pi} \left( \alpha\|A - P\tilde{A}P^T\|_2 + \beta\|X - P\tilde{X}\|_2 \right), \tag{1}$$

where $\Pi$ is the set of permutation matrices and $\alpha, \beta$ are hyper-parameters. Note that for un-attributed graphs, the distance in (1) aligns with the commonly used edit distance on graphs which is a measure of similarity between two graphs quantifying the minimal number of edges that need to be edited to convert one graph into another while taking into account graph isomorphism. Based on this distance, we can mathematically formulate the adversarial task as finding a perturbed attributed graph $(\tilde{G}, \tilde{X})$ within a specified budget $\epsilon$ such that $f(\tilde{G}, \tilde{X}) = \tilde{y} \neq y$ with $d^{\alpha,\beta}([G, X], [\tilde{G}, \tilde{X}]) < \epsilon$. Moreover, given that in practice the attacker does not have access to the ground-truth labels, we define an adversarial graph attack to be valid when $f(\tilde{G}, \tilde{X}) \neq f(G, X)$. We can now define the expected vulnerability of a graph function as its likelihood to suffer from such attacks in the input's neighborhood defined by $\epsilon$. Upper bounding this vulnerability allows us to quantify the model's expected robustness, we start by formulating the expected vulnerability of a graph function $f$ as

$$Adv_\epsilon^{\alpha,\beta}[f] = \mathbb{P}_{(G,X)\sim\mathcal{D}_{\mathcal{G},\mathcal{X}}}[(\tilde{G}, \tilde{X}) \in B^{\alpha,\beta}(G, X, \epsilon) : d_{\mathcal{Y}}(f(\tilde{G}, \tilde{X}), f(G, X)) > \sigma], \tag{2}$$

with $B^{\alpha,\beta}(G, X, \epsilon) = \{(\tilde{G}, \tilde{X}) : d^{\alpha,\beta}([G, X], [\tilde{G}, \tilde{X}]) < \epsilon\}$ for any budget $\epsilon \geq 0$. Additionally, $d_{\mathcal{Y}}$ can be any defined distance in the output space $\mathcal{Y}$ and $\sigma > 0$. Here, we focus on real-valued output mappings and consider the distance metric $d_{\mathcal{Y}}(f(\tilde{G}, \tilde{X}), f(G, X)) = \|f(\tilde{G}, \tilde{X}) - f(G, X)\|_{\mathcal{Y}}$. This formulation is applicable to both graph and node classification tasks. In node classification, the parameter $\sigma$ determines the minimal number of nodes that need to be successfully attacked to consider the attack to be adversarially successful at the graph-level. This allows for flexibility in different scenarios, where in some cases, even a single node's label flip may be considered a threat, while in others, a limited number of changes can be tolerated. In graph classification, the parameter $\sigma$ acts as a threshold to the continuous softmax output above which an attack is considered effective. We can now introduce the concept of expected robustness of a function defined on graphs, such as a GNN.

**Definition 3.1** (Expected Adversarial Robustness). Let $d^{\alpha,\beta}$ be a graph distance on the spaces $(\mathcal{G}, \mathcal{X})$ and $d_{\mathcal{Y}}$ be a distance on the space $\mathcal{Y}$. The graph function $f : (\mathcal{G}, \mathcal{X}) \to \mathcal{Y}$ is $((d^{\alpha,\beta}, \epsilon), (d_{\mathcal{Y}}, \gamma))-$ robust if its vulnerability as defined in (2) can be upper-bounded by $\gamma$, i.e., $Adv_\epsilon^{\alpha,\beta}[f] \leq \gamma$.

In Appendix A (Proposition A.1) we show how, via the equivalence of metrics, expected robustness in a given metric implies expected robustness in several other metrics.

**Relating Expected Adversarial Robustness to Worst-Case Adversarial Robustness.** Our introduced formulation represents a broader perspective of the classical "worst-case" adversarial robustness, where the attacker aims to identify a single adversarial attack representing "worst-case" losses within a predefined budget and neighborhood. Our Expected Adversarial Robustness focuses on understanding the overall behavior of the underlying graph-based function within the specified input neighborhood. This approach offers a more comprehensive assessment of the model's robustness. Nevertheless, we note that our formulation encompasses the adversarial robustness as a special case since by definition these examples are included in our considered neighborhood. In fact, by adjusting the hyper-parameter $\sigma$, we can isolate these worst-case examples. Lemma 3.2 directly relates Definition 3.1 to the classical "worst-case" adversarial robustness.

**Lemma 3.2.** *Let $d^{\alpha,\beta}$ be a defined graph metric on the metric spaces $\mathcal{G}, \mathcal{X}$. Let $f : (\mathcal{G}, \mathcal{X}) \to \mathcal{Y}$ be a graph-based function, we have the following result: If $f$ is $((d^{\alpha,\beta}, \epsilon), (d_{\mathcal{Y}}, \gamma))$–robust, then $f$ is also $((d^{\alpha,\beta}, \epsilon), (d_{\mathcal{Y}}, \gamma))$–"worst-case" robust.*

The proof of Lemma 3.2 is provided in Appendix B. As a result, our forthcoming theoretical analysis, which considers the general case, is equally applicable to worst-case adversarial examples, which will be also validated experimentally in Section 6. We finally note that the advantages and pitfalls of the generalization of "worst-case" to average robustness have also been studied by Rice et al. (2021).

## 4 THE EXPECTED ROBUSTNESS OF MESSAGE PASSING NEURAL NETWORKS

We now use our Expected Adversarial Robustness Definition 3.1 to derive an upper bound on the expected robustness of the GCN, based on which we introduce our more robust GCN adaptation.

### 4.1 ON THE EXPECTED ROBUSTNESS OF GRAPH CONVOLUTIONAL NETWORKS

Our work primarily focuses on the theoretical analysis of GCNs within the broader context of MPNNs. The computations of one GCN layer are composed of the aggregation of node hidden states over neighborhoods in the graph and subsequent node-wise updates of the hidden states via a weight matrix and non-linear activation function. The updated hidden states are then passed to the next layer for further aggregation and updates. An iteration of this process can be expressed as follows

$$h^{(\ell)} = \phi^{(\ell)}(\tilde{A} h^{(\ell-1)} W^{(\ell)}), \tag{3}$$

where $h^{(\ell)}$ represents the hidden state in the $\ell$-th GCN layer and $h^{(0)}$ is the initial node features $X \in \mathbb{R}^{n \times K}$, $W^{(\ell)} \in \mathbb{R}^{p \times e}$ is the weight matrix in the $\ell$-th layer, $e$ is the embedding dimension and $\phi^{(\ell)}$ is a 1-Lipschitz continuous non-linear activation function. Moreover, $\tilde{A} \in \mathbb{R}^{n \times n}$, with $n$ being the number the nodes, denotes the normalized adjacency matrix $\tilde{A} = D^{-1/2} A D^{-1/2}$.

Determining the exact expected adversarial robustness of a graph-based function, as outlined in Section 3, poses a significant challenge. To overcome this, we provide an upper bound, referred to as $\gamma$ in Definition 3.1. Our definition of adversarial attacks is closely related to the concept of sensitivity analysis. In both cases, the goal is to understand how small input changes can affect the model's output. We hence tackled the adversarial task by adopting an input perturbation perspective. Similar approaches based on sensitivity analysis have had some success for deep neural networks (DNNs). While extending to other domains such as Convolutional Neural Networks is direct, generalizing to graphs presents new challenges. Notably, model dynamics differ due to the Message Passing process involving the adjacency and node features, complicating the task of providing an upper-bound. Since, the architecture itself involves the adjacency matrix, any perturbation on the input produces a different dynamic in the model itself, unlike the classical DNNs architecture, which remains static when subject to perturbations. Consequently, the theoretical analysis and results have to reflect the underlying propagation architecture, i.e., the graph structure. Theorem 4.1 provides theoretical insight into the expected robustness of GCNs by establishing an upper bound on the amount of perturbation that a GCN can tolerate before its predictions become unreliable.

**Theorem 4.1.** *Let $f : (\mathcal{G}, \mathcal{X}) \to \mathcal{Y}$ denote a graph-based function composed of $L$ GCN layers, with $W^{(i)}$ denoting the weight matrix of the $i$-th layer. Further, let $d^{0,1}$ be a feature distance. For attacks targeting node features of the input graph, with a budget $\epsilon$, with respect to Definition 3.1:*

- $f$ is $((d^{0,1}, \epsilon), (d_1, \gamma))$–robust with $\gamma = \prod_{i=1}^{L} \|W^{(i)}\|_1 \epsilon (\sum_{u \in \mathcal{V}} \hat{w}_u)/\sigma$, with $\hat{w}_u$ denoting the sum of normalized walks of length $(L-1)$ starting from node $u$ and $\mathcal{V}$ is the node set.

- $f$ is $((d^{0,1}, \epsilon), (d_\infty, \gamma))$–robust with $\gamma = \prod_{i=1}^{L} \|W^{(i)}\|_\infty \epsilon \hat{w}_G/\sigma$, with $\hat{w}_G = \max_{u \in \mathcal{V}} \hat{w}_u$.

As previously mentioned, the provided upper-bound in Theorem 4.1 is directly dependent on both the graph structure (in terms of walks from the graph's nodes) and the propagation scheme (where the length of the considered walks depends on the number of message-passing iterations). The derived upper-bound is intuitive: effectively using feature-based attacks is increasingly difficult with increasing graph sparsity; or conversely, in dense graphs node feature attacks can have greater effect since the message passing scheme propagates them along a greater number of walks. To make this precise, the expected sum of normalized walks in a sparser graph tends to be lower than in a denser one, resulting in a reduced bound, indicating a more robust model within the considered neighborhood. We finally note that this bound applies to both targeted and untargeted feature modifications, whether limited to a specific subset or all the nodes. While our main focus is on node feature-based attacks, our analysis provided in Theorem 4.1 can be extended to structural attacks; Theorem 4.2 sheds light on this latter case.

**Theorem 4.2.** *Let $f : (\mathcal{G}, \mathcal{X}) \to \mathcal{Y}$ denote a graph function composed of $L$ GCN layers, where $W^{(i)}$ denotes the weight matrix of the $i$-th layer. Further, let $d^{1,0}$ be a graph distance. For structural attacks with a budget $\epsilon$, the function $f$ is $((d^{1,0}, \epsilon), (d_2, \gamma))$–robust with*

$$\gamma = \prod_{i=1}^{L} \|W^{(i)}\|_2 \|X\|_2 \epsilon (1 + L \prod_{i=1}^{L} \|W^{(i)}\|_2)/\sigma. \tag{4}$$

We observe the upper bound in (4) to functionally depend on the size of the graph via $\|X\|_2$, yielding the intuitive result that larger graphs have more potential targets to attack and thereby give rise to less robust models. The proofs of Theorems 4.1 and 4.2 are provided in Appendix C.

## 4.2 ON THE GENERALIZATION TO OTHER GRAPH NEURAL NETWORKS

While our work focuses on GCNs, our analysis can be extended to any GNN. Our theoretical analysis is contingent on assuming the input node feature space to be bounded, which is a realistic assumption for real word data. For illustration, Theorem 4.3 derives the upper-bound of the specific case of GINs.

**Theorem 4.3.** *Let $f : (\mathcal{G}, \mathcal{X}) \to \mathcal{Y}$ be composed of $L$ GIN-layers (with parameter $\zeta = 0$, that is usually denoted by $\epsilon$ in the literature) and $W^{(i)}$ denote the weight matrix of the $i$-th MLP layer. We consider the input node feature space to be bounded, i.e., $\|X\|_2 < B$ for some $B \in \mathbb{R}$, and graphs of maximum degree $\Delta_G$. For node feature-based attacks, with a budget $\epsilon$, the function $f$ is $((d^{0,1}, \epsilon), (d_\infty, \gamma))$–robust with*

$$\gamma = \prod_{l=1}^{L} \|W^{(l)}\|_\infty (B \, L \, \Delta_G + \epsilon)/\sigma.$$

Theorem 4.3 is proved in Appendix D. In addition, following the same assumption, it is possible to derive an upper bound on the GCN's robustness when subject to both structural and feature-based attacks simultaneously. The complete study and additional information are provided in Appendix F.

## 4.3 ENHANCING THE ROBUSTNESS OF GRAPH CONVOLUTIONAL NETWORKS

Leveraging the established upper-bound on the expected robustness in Theorem 4.1, we now introduce a novel approach, called Graph Convolutional Orthonormal Robust Networks (GCORNs), which enhances the robustness of a GCN to node feature perturbations while maintaining its ability to learn accurate node and graph representations. To this end, we aim to design a GCN architecture for which the upper bound $\gamma$ as stated in Theorem 4.1 is low. As this quantity is dependent on the norm of the weight matrices in each layer of the GCN, we propose to control the norms of these matrices.

Enhancing the robustness through enforcing matrix norm constraints during training has previously been studied in the context of DNNs through methods such as Parseval regularization (Cisse et al.,

2017) or optimizing over the orthogonal manifold (Ablin & Peyré, 2022). However, as observed in our experiments, the added constraints of these methods can negatively impact the performance of the graph function and additionally the hyper-parameter tuning can be tricky especially when dealing with large datasets. In our work, we choose to tackle the task by modifying the mathematical formulation of the GCN in (3) to encourage the orthonormality of the weights. We have therefore chosen to use an iterative algorithm (Björck & Bowie, 1971), that mainly was used in the literature for studying Lipschitz approximation (Anil et al., 2019), to ensure a fair trade-off between clean and attacked accuracy. We note that, based on the introduced Theorem 4.1, any orthonormalization method can theoretically enhance the underlying model's robustness. Given our weight matrix $W$, the iterative process which is computed using Taylor expansion, consists of finding the closest orthonormal matrix $\hat{W}$ to our weight matrix $W$. By considering $\hat{W}_0 = W$, we recursively compute $\hat{W}_k$ from

$$\hat{W}_{k+1} = \hat{W}_k \left( I + \tfrac{1}{2} Q_k + \ldots + (-1)^p \binom{-1/2}{p} Q_k^p \right), \tag{5}$$

with $k \geq 0$, $Q_k = I - \hat{W}_k^T \hat{W}_k$ and $p \geq 1$ is the chosen order. A key advantage of this orthonormalization approach is its differentiability, hence its compatibility with the back-propagation process of training GNNs. By incorporating this projection into each forward pass during the model's training, we encourage the orthonormality of the weights, consequently enhancing its expected robustness.

**Training and Convergence of Our Approach.** Encouraging the orthonormality of the weight matrices for each layer in our framework mitigates the problems of vanishing and exploding gradients. Our proposed method preserves the gradient norm, leading to enhanced convergence and improved learning (Guo et al., 2022). It is important to note that the training of our framework relies on the convergence of the iterative orthonormalization process. From the original work (Björck & Bowie, 1971) which examined this aspect, convergence is contingent on the condition $\|W^T W - I\| \leq 1$ being satisfied, which can be guaranteed by applying a scaling factor, based on the spectral norm, to the weight matrices before the iterative process. We noticed that this approach not only guarantees that the weight matrices satisfy the necessary condition for convergence but also helps to speed up the convergence and the training process as analyzed by previous work (Salimans & Kingma, 2016).

**Complexity of Our Approach.** Equation (5) entails a trade-off between convergence speed and the approximation's precision. A higher order $p$ yields closer projections in each iteration, resulting in an increased computational complexity. This trade-off must be carefully considered to strike a balance between complexity and approximation accuracy. The main complexity of the method results from the matrix products which are $\mathcal{O}(e^3)$ where $e$ represents the embedding dimension. Our experiments indicate that a low order and a small number of iterations often yield a satisfactory approximation in practice. We note that our method's complexity does not increase with growing input graph size, distinguishing it from other defenses such as GNNGuard which has a complexity of $\mathcal{O}(e \times |E|)$, where $|E|$ represents the number of edges or GNN-SVD which computes a low-rank approximation through SVD with a corresponding complexity of $\mathcal{O}(n^3)$, with $n$ being the number of nodes. We point out that we conducted an ablation study (reported in Appendix G) on the effect of changing the order $p$ and number of iterations $k$ on both the precision of the orthonormal projection (represented by both the clean accuracy and the adversarial accuracy) and the time complexity.

## 5 ESTIMATION OF OUR ROBUSTNESS MEASURE

Based on our introduced definition of expected robustness in Section 3, we introduce an algorithm to empirically estimate the quantity $Adv_\epsilon^{\alpha,\beta}[f]$, serving therefore as a new metric to evaluate GNN robustness. While most defense methods on graphs are evaluated using the worst-case accuracy of a GNN when exposed to specific attack schemes (Bojchevski et al., 2020; Xu et al., 2019a; Zügner & Günnemann, 2019), $Adv_\epsilon^{\alpha,\beta}[f]$ is an attack-independent and model-agnostic robustness metric based on uniform sampling. It can therefore be used as a *security checkpoint*, in addition to existing robustness metrics, to evaluate the GNN robustness against unknown attack distributions.

Numerous model-agnostic robustness metrics (Weng et al., 2018; Cheng et al., 2021) have emerged primarily in computer vision. These metrics mainly assess robustness through measuring the *distortion* between input and output manifolds. While these methods can be extended to the graph classification task with appropriate distortion definitions, their application in the node classification context proves challenging. To our knowledge, there exists no specific or analogous adaptation within the context of graph representation learning. Thus, our proposed evaluation aims to address this gap.

Given that the expected value of an indicator random variable for an event $E$ is the probability of that event, i.e., $\mathbb{E}[\mathbf{1}\{E\}] = P(E)$, we use (6) as an equivalent formulation of $Adv_\epsilon^{\alpha,\beta}[f]$ in (2).

$$Adv_\epsilon^{\alpha,\beta}[f] = \mathbb{E}_{\substack{(G,X)\sim\mathcal{D}_{\mathcal{G},\mathcal{X}}, \\ (\tilde{G},\tilde{X})\in B^{\alpha,\beta}((G,X),\epsilon)}} \left[\mathbf{1}\{d_{\mathcal{Y}}(f(\tilde{G},\tilde{X}), f(G,X)) > \sigma\}\right]. \tag{6}$$

From the law of large numbers (Bernoulli, 1713), the mean sampling is an unbiased estimator of the vulnerability quantity $Adv_\epsilon^{\alpha,\beta}[f]$. Consequently, we estimate a GNN's expected robustness by generating multiple graph pairs $([G,X],[\tilde{G},\tilde{X}])$, where each input graph $[G,X]$ is sampled from the underlying distribution $D_{\mathcal{G},\mathcal{X}}$ and $[\tilde{G},\tilde{X}]$ is uniformly sampled from the ball of radius $\epsilon$ centered around $[G,X]$, i.e., $d^{\alpha,\beta}([G,X],[\tilde{G},\tilde{X}]) \leq \epsilon$. This approach can be used for both structural and feature attacks. Since we focus on the latter, we propose a sampling strategy based on the distance

$$d^{0,1}([G,X],[\tilde{G},\tilde{X}]) = \|X - \tilde{X}\|_{\mathcal{X}} = \max_{i\in\{1,...,n\}}\|X_i - \tilde{X}_i\|_p, \tag{7}$$

where $\|\cdot\|_p$ is the $L_p$-norm ($p > 0$) and $X_i, \tilde{X}_i$ are the $i$-th rows of the matrices $X, \tilde{X} \in \mathbb{R}^{n\times K}$. Since the original dataset consists of samples $[G,X]$ from the underlying distribution $D_{\mathcal{G},\mathcal{X}}$, we only need to sample uniformly graphs $[\tilde{G},\tilde{X}]$ from the neighborhood of the dataset graphs $[G,X]$ such that $\|X - \tilde{X}\|_{\mathcal{X}} \leq \epsilon$ and $\tilde{G} = G$. Sampling such $\tilde{X}$ is equivalent to first sample $Z \in \mathbb{R}^{n\times K}$ from $\mathcal{B}_\epsilon = \{Z \in \mathbb{R}^{n\times K} : \|Z\|_{\mathcal{X}} \leq \epsilon\}$ and then set $\tilde{X} = X + Z$. Sampling from the ball $\mathcal{B}_\epsilon$ could be performed by partitioning the ball with respect to the possible values of $\|Z\|_{\mathcal{X}}$.

$$S_r = \{Z \in \mathbb{R}^{n\times K} : \|Z\|_{\mathcal{X}} = r\}, \qquad \mathcal{B}_\epsilon = \cup_{r\leq\epsilon}S_r; \qquad \forall r \neq r' \quad S_r \cap S_{r'} = \emptyset.$$

We therefore propose to use the following *Stratified Sampling* approach, which is based on two main steps: **(i)** Sample a distance $r$ from $[0,\epsilon]$ using the distribution $p_\epsilon$ defined in Lemma 5.1; **(ii)** Sample $Z$ from $S_r$. Additional details on the prior distribution $r$ and the sampling from $S_r$ are in Appendix H.

**Lemma 5.1.** *Let $\mathbb{R}^K$ be the real finite-dimensional space and $\epsilon$ a positive real number. If $R^{(p)}$ is the random variable indicating the maximum of the $L_p$ norm's values inside the ball of radius $\epsilon$, i.e., $\mathcal{B}_\epsilon = \{Z \in \mathbb{R}^{n\times K} : \max_{i\in\{1,...,n\}}\|Z_i\|_p \leq \epsilon\}$. Then, for every $p > 0$, the density distribution of $R^{(p)}$ does not depends on $p$ and is defined as follows, $p_\epsilon(r) = K\frac{1}{\epsilon}\left(\frac{r}{\epsilon}\right)^{K-1}\mathbf{1}\{0 \leq r \leq \epsilon\}$.*

The proof is available in Appendix I. We note that the proposed framework is a practical and theory-based robustness evaluation approach applicable to any GNN, since no assumption has been made on the underlying architecture. Algorithm 1 in Appendix H offers a summary of the approach.

## 6 EMPIRICAL INVESTIGATION

This section examines the practical impact of our theoretical findings on real-world datasets where we aim to investigate the robustness of GCORN in comparison to other defense benchmarks.

### 6.1 EXPERIMENTAL SETUP

The necessary code to reproduce all our experiments is available on github https://github.com/Sennadir/GCORN. In this section, we focus on node classification while we report results on graph classification (Morris et al., 2020) in Appendix J. We use the citations networks Cora, CiteSeer, and PubMed (Sen et al., 2008), the Co-author network CS (Shchur et al., 2018), and the citation network between Computer Science arXiv papers OGBN-Arxiv (Hu et al., 2020). Further information about the datasets and implementation details are provided in Appendix L. To reduce the impact of initialization, we repeated each experiment 10 times and used the train/validation/test splits provided with the datasets and for the CS dataset, we followed the framework of Yang et al. (2016). We note that for the node feature-based attacks, we normalized the input features to work in a continuous space allowing more flexibility in terms of available attacks.

**Attacks.** We use two evasion and one poisoning feature-based attacks: **(i)** The baseline random attack injecting Gaussian noise $\mathcal{N}(0,\mathbf{I})$ to the features with a scaling parameter $\psi$ controlling the attack budget; **(ii)** The white-box Proximal Gradient Descent (Xu et al., 2019a), which is a gradient-based

Table 1: Attacked classification accuracy ($\pm$ standard deviation) of the models on different benchmark node classification dataset after the feature attacks.

| Attack | Dataset | GCN | GCN-k | AirGNN | RGCN | ParsevalR | GCORN |
|---|---|---|---|---|---|---|---|
| Random ($\psi = 0.5$) | Cora | $68.4 \pm 1.9$ | $69.2 \pm 2.6$ | $73.5 \pm 1.9$ | $71.6 \pm 0.3$ | $72.9 \pm 0.9$ | $\mathbf{77.1 \pm 1.8}$ |
| | CiteSeer | $57.8 \pm 1.5$ | $62.3 \pm 1.2$ | $64.6 \pm 1.6$ | $63.7 \pm 0.6$ | $65.1 \pm 0.8$ | $\mathbf{67.8 \pm 1.4}$ |
| | PubMed | $68.3 \pm 1.2$ | $71.2 \pm 1.1$ | $70.9 \pm 1.3$ | $71.4 \pm 0.5$ | $71.8 \pm 0.8$ | $\mathbf{73.1 \pm 1.1}$ |
| | CS | $85.3 \pm 1.1$ | $86.7 \pm 1.1$ | $87.5 \pm 1.6$ | $88.2 \pm 0.9$ | $87.6 \pm 0.6$ | $\mathbf{89.8 \pm 1.2}$ |
| | OGBN-Arxiv | $68.2 \pm 1.5$ | $52.8 \pm 0.5$ | $66.5 \pm 1.3$ | $63.8 \pm 1.9$ | $68.3 \pm 1.9$ | $\mathbf{69.1 \pm 1.8}$ |
| Random ($\psi = 1.0$) | Cora | $41.7 \pm 2.1$ | $46.3 \pm 2.8$ | $53.7 \pm 2.2$ | $52.8 \pm 1.6$ | $55.3 \pm 1.2$ | $\mathbf{57.6 \pm 1.9}$ |
| | CiteSeer | $38.2 \pm 1.3$ | $45.3 \pm 1.4$ | $49.8 \pm 2.1$ | $43.7 \pm 2.2$ | $51.2 \pm 1.2$ | $\mathbf{57.3 \pm 1.7}$ |
| | PubMed | $60.1 \pm 1.7$ | $62.3 \pm 1.3$ | $62.4 \pm 1.2$ | $61.9 \pm 1.2$ | $61.3 \pm 1.7$ | $\mathbf{65.8 \pm 1.4}$ |
| | CS | $69.9 \pm 1.3$ | $73.2 \pm 0.9$ | $76.7 \pm 2.8$ | $76.2 \pm 1.4$ | $78.7 \pm 1.2$ | $\mathbf{81.3 \pm 1.6}$ |
| | OGBN-Arxiv | $66.4 \pm 1.9$ | $46.6 \pm 0.6$ | $62.7 \pm 1.6$ | $63.0 \pm 2.4$ | $66.1 \pm 0.7$ | $\mathbf{67.3 \pm 2.1}$ |
| PGD | Cora | $54.1 \pm 2.4$ | $58.3 \pm 1.6$ | $68.2 \pm 1.8$ | $62.5 \pm 1.2$ | $68.6 \pm 1.7$ | $\mathbf{71.1 \pm 1.4}$ |
| | CiteSeer | $52.3 \pm 1.1$ | $59.6 \pm 1.6$ | $59.3 \pm 2.1$ | $61.9 \pm 1.1$ | $62.1 \pm 1.5$ | $\mathbf{65.6 \pm 1.4}$ |
| | PubMed | $66.1 \pm 2.1$ | $67.3 \pm 1.3$ | $70.8 \pm 1.7$ | $69.5 \pm 0.9$ | $68.9 \pm 2.1$ | $\mathbf{72.3 \pm 1.3}$ |
| | CS | $71.3 \pm 1.1$ | $74.1 \pm 0.8$ | $76.3 \pm 2.1$ | $76.6 \pm 1.2$ | $77.3 \pm 0.6$ | $\mathbf{79.6 \pm 1.2}$ |
| | OGBN-Arxiv | $67.5 \pm 0.9$ | $49.9 \pm 0.7$ | $55.7 \pm 0.9$ | $63.6 \pm 0.7$ | $67.6 \pm 1.2$ | $\mathbf{68.1 \pm 1.1}$ |
| Nettack | Cora | $60.9 \pm 2.5$ | $64.2 \pm 5.2$ | $66.7 \pm 3.8$ | $63.4 \pm 3.8$ | $67.5 \pm 2.5$ | $\mathbf{68.3 \pm 1.4}$ |
| | CiteSeer | $55.8 \pm 1.4$ | $71.7 \pm 1.4$ | $67.5 \pm 2.5$ | $70.8 \pm 3.8$ | $69.2 \pm 3.8$ | $\mathbf{77.5 \pm 2.5}$ |
| | PubMed | $60.0 \pm 2.5$ | $65.8 \pm 2.9$ | $69.2 \pm 1.4$ | $\mathbf{71.7 \pm 3.8}$ | $68.3 \pm 1.4$ | $70.8 \pm 1.4$ |
| | CS | $55.8 \pm 1.4$ | $71.6 \pm 1.4$ | $76.7 \pm 1.4$ | $71.7 \pm 2.9$ | $75.8 \pm 2.8$ | $\mathbf{78.3 \pm 1.4}$ |
| | OGBN-Arxiv | $49.2 \pm 2.9$ | $53.3 \pm 1.4$ | $\mathbf{56.7 \pm 1.4}$ | $52.6 \pm 2.5$ | $55.8 \pm 1.4$ | $55.8 \pm 1.4$ |

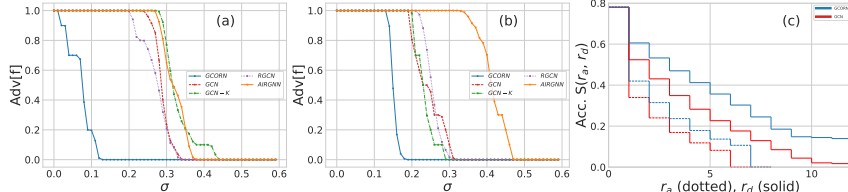

Figure 1: (a) and (b) display $Adv_\epsilon^{\alpha,\beta}[f]$ for Cora and OGBN-Arxiv. (c) Robustness guarantees on Cora, where $r_a, r_d$ are respectively the maximum number of adversarial additions and deletions.

approach to the adversarial optimization task. We set the perturbation rate to 15%. While this attack has limited success for structural perturbations due to the discrete space, it is known to be powerful in continuous spaces, such as the node feature space; **(iii)** The targeted-attack Nettack (Zügner et al., 2018), where similar to the original study, we selected 40 correctly classified target nodes, comprising 10 with the largest classification margin, 20 randomly chosen, and 10 with the smallest margin.

**Baseline Models.** Currently, as presented in Section 2, there are only few methods for defending against feature-based adversarial perturbations. We compare against **(i)** GCN-k (Seddik et al., 2022); **(ii)** RobustGCN (RGCN) (Zhu et al., 2019); **(iii)** AIRGNN (Liu et al., 2021a) and **(iv)** we finally compared to the Parseval Regularization (ParsevalR) (Cisse et al., 2017), another orthonormalization method, with great success in the computer vision. The aim is mainly to motivate the merits of our proposed orthonormalization method based on the iterative weight projection.

## 6.2 EXPERIMENTAL RESULTS

**Empirical Estimation of Our Expected Robustness.** We start by investigating the robustness of the different considered benchmarks through the empirical estimation of $Adv_\epsilon^{\alpha,\beta}[f]$ as introduced in Section 5 (see Appendix M and L). Figure 1 (a) and (b) report the estimated values of the vulnerability $Adv_\epsilon^{\alpha,\beta}[f]$ for our GCORN and the considered defense benchmarks for the Cora and OGBN-Arxiv datasets, respectively. Additional results on further datasets are provided in Appendix J. The figures show that our GCORN yields the lowest adversarial vulnerability indicating therefore that our proposed approach is exhibiting greater expected robustness within the considered neighborhood.

Table 2: Attacked classification accuracy ($\pm$ standard deviation) of the models on different benchmark node classification datasets after the structural attacks.

| Attack | Dataset | GCN | GCN-Jaccard | RGCN | GNN-SVD | GNN-Guard | ParsevalR | GCORN |
|--------|---------|-----|-------------|------|---------|-----------|-----------|-------|
| Mettack | Cora | $73.0 \pm 0.7$ | $75.4 \pm 1.8$ | $69.2 \pm 0.3$ | $73.6 \pm 0.9$ | $74.4 \pm 0.8$ | $71.9 \pm 0.7$ | $\mathbf{77.3 \pm 0.5}$ |
| | CiteSeer | $63.2 \pm 0.9$ | $69.5 \pm 1.9$ | $68.9 \pm 0.6$ | $65.8 \pm 0.6$ | $68.8 \pm 1.5$ | $68.3 \pm 0.8$ | $\mathbf{73.7 \pm 0.3}$ |
| | PubMed | $60.7 \pm 0.7$ | $62.9 \pm 1.8$ | $65.1 \pm 0.4$ | $82.1 \pm 0.8$ | $\mathbf{84.8 \pm 0.3}$ | $69.5 \pm 1.1$ | $71.8 \pm 0.4$ |
| | CoraML | $73.1 \pm 0.6$ | $75.4 \pm 0.4$ | $77.1 \pm 1.1$ | $71.3 \pm 1.0$ | $76.5 \pm 0.7$ | $76.9 \pm 1.3$ | $\mathbf{79.2 \pm 0.6}$ |
| PGD | Cora | $76.7 \pm 0.9$ | $78.3 \pm 1.1$ | $72.0 \pm 0.3$ | $71.6 \pm 0.4$ | $75.0 \pm 2.0$ | $78.4 \pm 1.2$ | $\mathbf{79.9 \pm 0.4}$ |
| | CiteSeer | $67.8 \pm 0.8$ | $70.9 \pm 1.0$ | $62.2 \pm 1.8$ | $60.3 \pm 2.4$ | $68.9 \pm 2.2$ | $70.6 \pm 1.0$ | $\mathbf{73.1 \pm 0.5}$ |
| | PubMed | $75.3 \pm 1.6$ | $73.8 \pm 1.3$ | $78.6 \pm 0.4$ | $81.9 \pm 0.4$ | $\mathbf{84.3 \pm 0.4}$ | $77.3 \pm 0.7$ | $77.4 \pm 0.4$ |
| | CoraML | $76.9 \pm 1.2$ | $75.0 \pm 2.4$ | $77.5 \pm 0.3$ | $73.1 \pm 0.5$ | $75.5 \pm 0.8$ | $81.3 \pm 0.4$ | $\mathbf{84.1 \pm 0.2}$ |
| DICE | Cora | $74.9 \pm 0.8$ | $76.9 \pm 0.9$ | $79.6 \pm 0.3$ | $72.2 \pm 1.4$ | $75.6 \pm 1.1$ | $\mathbf{79.7 \pm 0.8}$ | $78.9 \pm 0.4$ |
| | CiteSeer | $64.1 \pm 0.5$ | $66.0 \pm 0.6$ | $68.7 \pm 0.5$ | $62.6 \pm 1.2$ | $65.5 \pm 1.1$ | $68.9 \pm 0.4$ | $\mathbf{74.6 \pm 0.4}$ |
| | PubMed | $79.4 \pm 0.4$ | $78.3 \pm 0.2$ | $\mathbf{79.8 \pm 0.4}$ | $76.6 \pm 0.5$ | $77.8 \pm 0.7$ | $79.2 \pm 0.3$ | $78.1 \pm 0.6$ |
| | CoraML | $78.3 \pm 0.6$ | $77.5 \pm 0.3$ | $80.1 \pm 0.4$ | $58.7 \pm 0.4$ | $77.5 \pm 0.2$ | $80.5 \pm 1.3$ | $\mathbf{81.1 \pm 0.8}$ |

**Worst-Case Adversarial Evaluation.** Table 1 reports the attacked node classification accuracies for our GCORN and the other considered benchmarks. The results show that, when subject to features-based adversarial attacks with a varying budget, the performance of GCN is severely impacted. Instead, the proposed GCORN demonstrates a significant improvement in defending against these attacks in comparison to other methods. For instance, GCORN is outperforming the GCN-k by an average of $\sim 12\%$ in accuracy. In certain situations, GCORN demonstrates the ability to recover the GCN's performance to a level equivalent to no adversarial attack. Furthermore, the baselines are not as effective in defending against gradient-based attacks. In contrast, GCORN, which is based on an attack-agnostic upper bound (Theorem 4.1), demonstrates its *ability to defend against gradient-based attacks*. We, additionally study the trade-off between robustness and input sensitivity in Appendix J.

**Certified Robustness.** In addition to our probabilistic and empirical evaluations, we assessed the certified robustness of a GCN and our proposed GCORN on the Cora dataset using the sparse randomized smoothing approach (Bojchevski et al., 2020). Specifically, we plotted their certified accuracy $S(r_a, r_d)$ for varying addition $r_a$ and deletion $r_d$ radii. Figure 1(c) showcases our theoretical robustness; GCORN improves the certified accuracy with respect to feature perturbations for all radii.

**Structural Perturbations.** Although our focus is on node feature-based adversarial attacks, we also provided a theoretical analysis for structural perturbations (Theorem 4.2). To evaluate the effectiveness of our approach in handling structural attacks, we conducted an empirical comparison to five baseline defense methods that focus on structural perturbations, GNN-Jaccard (Wu et al., 2019), RobustGCN (Zhu et al., 2019), GNN-SVD (Entezari et al., 2020), GNNGuard (Zhang & Zitnik, 2020) and the Parseval Regularization(ParsevalR) (Cisse et al., 2017). We considered three structural adversarial attacks with a perturbation budget $0.1|\mathcal{E}|$: "Mettack" (with the 'Meta-Self' strategy) (Zügner & Günnemann, 2019), "Dice" (Zügner & Günnemann, 2019) and "PGD" (Xu et al., 2019a). The average node classification accuracies of the considered models under attack are presented in Table 2. Our GCORN shows remarkable defense capabilities against these attacks, outperforming other methods in 8 of 12 experiments. This result highlights the effectiveness of our proposed defense mechanism in enhancing the robustness of the underlying GCNs against structural perturbations while providing theoretical guarantees which are absent for the considered baselines.

# 7 CONCLUSION

We have proposed GCORN, an adaptation of the GCN, which is robust against feature-based adversarial attacks. The approach utilizes the orthonormalization of weight matrices to control the robustness of GCNs via an upper bound we derived. Additionally, we have proposed a probabilistic evaluation method for the robustness of GNNs based on the introduced definition of "Expected Adversarial Robustness". Experimental results comparing our GCORN model to the standard GCN and existing defense methods show the superior performance of GCORN on different real-world datasets.

## ACKNOWLEDGEMENTS

This work was partially supported by the Wallenberg AI, Autonomous Systems and Software Program (WASP) funded by the Knut and Alice Wallenberg Foundation. The computation (on GPUs) was enabled by resources provided by the National Academic Infrastructure for Supercomputing in Sweden (NAISS) at Alvis partially funded by the Swedish Research Council through grant agreement no. "2022-06725". This work was granted access to the HPC resources of IDRIS under the allocation "2023-AD010613410R1" made by GENCI. Y.A. is supported by the French National research agency via the AML-HELAS (ANR-19-CHIA-0020) project. We furthermore want to thank Dr. Henrique Helfer Hoeltgebaum for a very helpful discussion on cybersecurity during the rebuttal period.

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

# Appendix: Bounding the Expected Robustness of Graph Neural Networks Subject to Node Feature Attacks

## A  PROOF OF PROPOSITION A.1

**Proposition A.1.** *Let $f : (\mathcal{G}, \mathcal{X}) \to \mathcal{Y}$ be a graph-based classifier subject to feature-based attacks and $\mathcal{X}$ be of dimension $D$. Let $d_2^{0,1}$ be the graph distance corresponding to the spectral norm, $d_1^{0,1}$ corresponding to the $L_1$ norm and $d_p^{0,1}$ corresponding to the $L_p$ norm with $p > 2$. If $f$ is $((d_2^{0,1}, \epsilon), (d_\mathcal{Y}, \gamma))$–robust, then $f$ is also $((d_p^{0,1}, D^{-1/2}\epsilon), (d_\mathcal{Y}, \gamma))$–robust and $((d_1^{0,1}, D^{-2}\epsilon), (d_\mathcal{Y}, \sqrt{D}\gamma))$–robust.*

*Proof.* Let $\mathbb{R}^D$ the real finite-dimensional space. The set of standard considered norms of $\mathbb{R}^D$ for $x = (x_1, \ldots, x_D) \in \mathbb{R}^D$ is

$$\forall p > 0, \ \|x\|_p = \left( \sum_{i=1}^{D} |x_i|^p \right)^{1/p}, \qquad \|x\|_\infty = \max_{1 \le i \le n} |x_i|.$$

As for any $x \in \mathbb{R}^D$, the mapping $p \mapsto \|x\|_p$ is monotone decreasing (Raıssouli & Jebril, 2010) , meaning that

$$q > p \ge 1 \Rightarrow \|x\|_q \le \|x\|_p.$$

Using *Holder's inequality* with $s = q/p > 1$, we have

$$
\begin{aligned}
(\|x\|_p)^p &= \sum_{i=1}^{D} |x_i|^p \\
&= \sum_{i=1}^{D} |x_i|^p \cdot 1 \\
&\le \left( \sum_{i=1}^{D} (|x_i|^p)^s \right)^{\frac{1}{s}} \left( \sum_{i=1}^{D} 1^{\frac{s}{s-1}} \right)^{1-\frac{1}{s}} \\
&= \left( \sum_{i=1}^{D} (|x_i|^p)^{q/p} \right)^{\frac{p}{q}} \left( \sum_{i=1}^{D} 1^{\frac{q}{q-p}} \right)^{1-\frac{p}{q}} \\
&= D^{1-p/q} (\|x\|_q)^p.
\end{aligned}
$$

Thus,

$$q > p \ge 1 \Rightarrow \|x\|_p \le D^{1/p - 1/q} \|x\|_q. \tag{8}$$

We additionally have that

$$
\begin{aligned}
\|x\|_q &= \left( \sum_{i=1}^{D} |x_i|^q \right)^{1/q}, \\
&\le \left( \sum_{i=1}^{D} \|x\|_\infty^q \right)^{1/q} \\
&\le D^{1/q} \|x\|_\infty \\
&\le D^{1/q} \|x\|_p. \tag{9}
\end{aligned}
$$

From (8) and (9), we deduce that

$$q > p \geq 1 \Rightarrow D^{\frac{1}{q} - \frac{1}{p}} \|x\|_p \leq \|x\|_q \leq D^{\frac{1}{q}} \|x\|_p,$$

Consequently, for any matrix $N \in \mathbb{R}^{n \times D}$, we have the following

$$p > q \geq 1 \Rightarrow \forall x \neq 0 \begin{cases} D^{\frac{1}{q} - \frac{1}{p}} \|Nx\|_p \leq \|Nx\|_q \leq D^{\frac{1}{q}} \|Nx\|_p, \\ D^{\frac{1}{q} - \frac{1}{p}} \|x\|_p \leq \|x\|_q \leq D^{\frac{1}{q}} \|x\|_p, \end{cases}$$

$$\Rightarrow \forall x \neq 0 \begin{cases} D^{\frac{1}{q} - \frac{1}{p}} \|Nx\|_p \leq \|Nx\|_q \leq D^{\frac{1}{q}} \|Nx\|_p, \\ D^{-\frac{1}{q}} \frac{1}{\|x\|_p} \leq \frac{1}{\|x\|_q} \leq D^{-\frac{1}{q} + \frac{1}{p}} \frac{1}{\|x\|_p}, \end{cases}$$

$$\Rightarrow \forall x \neq 0, \ D^{-\frac{1}{p}} \frac{\|Nx\|_p}{\|x\|_p} \leq \frac{\|Nx\|_q}{\|x\|_q} \leq D^{\frac{1}{p}} \frac{\|Nx\|_p}{\|x\|_p},$$

$$\Rightarrow \ D^{-\frac{1}{p}} \|N\|_p \leq \|N\|_q \leq D^{\frac{1}{p}} \|N\|_p.$$

In particular, for $p = 2$, we have

$$\forall q > 2, \forall N \in \mathbb{R}^{n \times D} \quad \|N\|_2 \leq \sqrt{D} \|N\|_q. \tag{10}$$

Similar for $q = 2$ and $p = 1$, we have

$$\forall N \in \mathbb{R}^{n \times D} \quad \|N\|_2 \leq D \|N\|_1. \tag{11}$$

The Inequalities (10) and (11) stay valid for the distance related to the norm $L_p$, i. e.,

$$\forall X, \tilde{X} \in \mathbb{R}^{n \times D}, \forall p > 2, \quad d_2(X, \tilde{X}) \leq \sqrt{D} d_p(X, \tilde{X}),$$

$$d_2(X, \tilde{X}) \leq D d_1(X, \tilde{X}).$$

Finally, for every input graph $G, X$, we have

$$\forall p > 2, \ \{(G', X') \mid d_p([G, X], [\tilde{G}, \tilde{X}]) < \frac{1}{\sqrt{D}} \epsilon\} \subset \{(G', X') \mid d_2([G, X], [\tilde{G}, \tilde{X}]) < \epsilon\},$$

$$\{(G', X') \mid d_1([G, X], [\tilde{G}, \tilde{X}]) < \frac{1}{D} \epsilon\} \subset \{(G', X') \mid d_1([G, X], [\tilde{G}, \tilde{X}]) < \epsilon\}.$$

Therefore, the $((d_2^{0,1}, \epsilon), (d_{\mathcal{Y}}, \gamma))$-robustness of a graph function $f$ implies the $((d_p^{0,1}, D^{-1/2}\epsilon), (d_{\mathcal{Y}}, \gamma))$–robustness and $((d_1^{0,1}, D^{-1}\epsilon), (d_{\mathcal{Y}}, \sqrt{D}\gamma))$–robustness.

$\square$

# B  PROOF OF LEMMA 3.2

**Lemma.** Let $d^{\alpha,\beta}$ be a defined graph metric on the metric spaces $\mathcal{G}, \mathcal{X}$. Let $f : (\mathcal{G}, \mathcal{X}) \to \mathcal{Y}$ be a graph-based function, we have the following result: If $f$ is $((d^{\alpha,\beta}, \epsilon), (d_{\mathcal{Y}}, \gamma))$–robust, then $f$ is also $((d^{\alpha,\beta}, \epsilon), (d_{\mathcal{Y}}, \gamma))$–"worst-case" robust.

*Proof.* Let $\mathcal{W}_\epsilon^{\alpha,\beta} = \{(A_w, X_w); d^{\alpha,\beta}([G, X], [\tilde{G}, \tilde{X}]) \leq \epsilon\}$ be the set of "worst-case" adversarial attacks within the attack budget $\epsilon$. We denote the expected vulnerability of a graph function $f$ on the set of worst-case adversarial examples as $Adv_\epsilon^{\alpha,\beta}[f]|_{\mathcal{W}_\epsilon^{\alpha,\beta}}$.

By definition, we have $\mathcal{W}_\epsilon^{\alpha,\beta} \subset B^{\alpha,\beta}(G, X, \epsilon)$. Consequently, we have that

$$Adv_\epsilon^{\alpha,\beta}[f]|_{\mathcal{W}_\epsilon^{\alpha,\beta}} = \mathbb{P}_{(G,X)\sim\mathcal{D}_{\mathcal{G},\mathcal{X}}}[(\tilde{G}, \tilde{X}) \in \mathcal{W}_\epsilon^{\alpha,\beta} : d_{\mathcal{Y}}(f(\tilde{G}, \tilde{X}), f(G, X)) > \sigma] \tag{12}$$

$$\leq \mathbb{P}_{(G,X)\sim\mathcal{D}_{\mathcal{G},\mathcal{X}}}[(\tilde{G}, \tilde{X}) \in B^{\alpha,\beta}(G, X, \epsilon) : d_{\mathcal{Y}}(f(\tilde{G}, \tilde{X}), f(G, X)) > \sigma] \tag{13}$$

$$\leq Adv_\epsilon^{\alpha,\beta}[f]. \tag{14}$$

Let the graph function $f$ be $((d^{\alpha,\beta}, \epsilon), (d_{\mathcal{Y}}, \gamma))$–robust, then $Adv_\epsilon^{\alpha,\beta}[f] \leq \gamma$. From (14), we have

$$\mathcal{W} - Adv_\epsilon^{\alpha,\beta}[f] \leq Adv_\epsilon^{\alpha,\beta}[f] \leq \gamma.$$

We therefore, can conclude that if $f$ is $((d^{\alpha,\beta}, \epsilon), (d_{\mathcal{Y}}, \gamma))$–robust, then $f$ is $((d^{\alpha,\beta}, \epsilon), (d_{\mathcal{Y}}, \gamma)) -$ "worst-case" robust.

$\square$

## C   PROOF OF THEOREM 4.1 AND 4.2

**Theorem.** *Let* $f : (\mathcal{G}, \mathcal{X}) \to \mathcal{Y}$ *denote a graph-based function composed of $L$ GCN layers, with* $W^{(i)}$ *denoting the weight matrix of the $i$-th layer. Further, let $d^{0,1}$ be a feature distance. For attacks targeting node features of the input graph, with a budget $\epsilon$, with respect to Definition 3.1:*

- *$f$ is $((d^{0,1}, \epsilon), (d_1, \gamma))$–robust with $\gamma = \prod_{i=1}^{L} \|W^{(i)}\|_1 \epsilon (\sum_{u \in \mathcal{V}} \hat{w}_u)/\sigma$, with $\hat{w}_u$ denoting the sum of normalized walks of length $(L-1)$ starting from node $u$ and $\mathcal{V}$ is the node set.*

- *$f$ is $((d^{0,1}, \epsilon), (d_\infty, \gamma))$–robust with $\gamma = \prod_{i=1}^{L} \|W^{(i)}\|_\infty \epsilon \hat{w}_G/\sigma$, with $\hat{w}_G = \max_{u \in \mathcal{V}} \hat{w}_u$.*

*Proof.* Let's consider a graph-function $f$ that is based on $L$ layers of GCN. The GCN message-passing propagation can be written for a node $u$ as

$$h_u^{(\ell)} = \sigma^{(\ell)}\Big( \sum_{v \in \mathcal{N}(u) \bigcup \{u\}} \frac{W^{(\ell)} h_v^{(\ell-1)}}{\sqrt{(1+d_u)(1+d_v)}} \Big) \tag{15}$$

where $W^{(\ell)} \in \mathbb{R}^{d_{\ell-1} \times d_\ell}$ is the learnable weight matrix with $d_\ell$ being the embedding dimension of layer $\ell$ and $\sigma^{(\ell)}$ is the activation function of $\ell$-th layer. We recall that $h^{(0)} = X \in \mathbb{R}^{n \times d}$ is set to the initial node features.

Let $X$ the original node features and denote by $X'$ the perturbed adversarial features. Let's consider a node $u \in V$, we denote by $h_u$ its representation in the clean graph and $h'_u$ its representation in the attacked graph , we note that the activation functions $(\sigma^{(\ell)})_{1 \leq \ell \leq L}$ are *nonexpensive*. Since we only consider node feature based adversarial attacks, then by definition the original node $u$ and its corresponding node in the corrupted graph have the same neighborhood, we can therefore write

$$\|h_u^{(L)} - h'^{(L)}_u\| = \|h_u^{(\ell)} - h'^{(\ell)}_u\|$$

$$= \|\sigma^{(\ell)}\Big( \sum_{v \in \mathcal{N}(u) \bigcup \{u\}} \frac{W^{(\ell)} h_v^{(\ell-1)}}{\sqrt{(1+d_u)(1+d_v)}} \Big) - \sigma^{(\ell)}\Big( \sum_{v' \in \mathcal{N}(u) \bigcup \{u\}} \frac{W^{(\ell)} h'^{(\ell-1)}_{v'}}{\sqrt{(1+d_u)(1+d_{v'})}} \Big)\|$$

$$\leq \|W^{(\ell)}\| \| \sum_{v \in \mathcal{N}(u) \bigcup \{u\}} \frac{h_v^{(\ell-1)}}{\sqrt{(1+d_u)(1+d_v)}} - \sum_{v' \in \mathcal{N}(u) \bigcup \{u\}} \frac{h'^{(\ell-1)}_{v'}}{\sqrt{(1+d_u)(1+d_{v'})}}\|$$

$$\leq \|W^{(\ell)}\| \| \sum_{v \in \mathcal{N}(u) \bigcup \{u\}} \frac{h_v^{(\ell-1)} - h'^{(\ell-1)}_v}{\sqrt{(1+d_u)(1+d_v)}}\|$$

$$= \|W^{(\ell)}\| \| \sum_{v \in \mathcal{N}(u) \bigcup \{u\}} \frac{1}{\sqrt{(1+d_u)(1+d_v)}} [\sigma^{(\ell-1)}\Big( \sum_{j \in \mathcal{N}(v) \bigcup \{v\}} \frac{W^{(\ell-1)} h_j^{(\ell-2)}}{\sqrt{(1+d_v)(1+d_j)}} \Big) -$$

$$\sigma^{(\ell-1)}\Big( \sum_{j \in \mathcal{N}(v) \bigcup \{v\}} \frac{W^{(\ell-1)} h'^{(\ell-2)}_j}{\sqrt{(1+d_v)(1+d_j)}} \Big)]\|$$

$$\leq \|W^{(\ell)}\| \|W^{(\ell-1)}\| \| \sum_{v \in \mathcal{N}(u) \bigcup \{u\}} \frac{1}{\sqrt{(1+d_u)(1+d_v)}} \sum_{j \in \mathcal{N}(v) \bigcup \{v\}} \frac{h_j^{(\ell-1)} - h'^{(\ell-1)}_j}{\sqrt{(1+d_v)(1+d_j)}}\|$$

By iteration on the same process, we get the following result:

$$\|h_u^{(L)} - h'^{(L)}_{u'}\| \le \prod_{l=1}^{L}\|W^{(l)}\|_2\| \underset{v\in\mathcal{N}(u)\bigcup\{u\}}{\Sigma}\underset{j\in\mathcal{N}(v)\bigcup\{v\}}{\Sigma}\cdots$$

$$\underset{z\in\mathcal{N}(y)\bigcup\{y\}}{\Sigma}\frac{X_u - X'_u}{\sqrt{(1+d_u)}(1+d_w)(1+d_j)\ldots(1+d_y)\sqrt{(1+d_z)}}\|$$

$$\le \prod_{l=1}^{L}\|W^{(l)}\|\|\hat{w}_u\|\epsilon$$

with $\hat{w}_u$ being the sum of normalized walks of length $(L-1)$ starting from node $u$.

Let's now consider the final output of the model which represents in the case of node classification the individual output of each node.

$$\|f(A,X) - f(A,X')\| = \|\begin{bmatrix} \vdots \\ h_u^{(L)} - h'^{(L)}_u \\ \vdots \end{bmatrix}\|$$

Based on the previous formulation, we have the following results:

$$\|f(A,X) - f(A,X')\|_1 \le \epsilon\prod_{l=1}^{L}\|W^{(l)}\|_1\sum_{u\in\mathcal{V}}\hat{w}_u$$

$$\|f(A,X) - f(A,X')\|_\infty = \epsilon\prod_{l=1}^{L}\|W^{(l)}\|_\infty\max_{u\in\mathcal{V}}\hat{w}_u$$

Therefore, we computed the Lipschitz constant of our considered model $f$. To link this constant to our robustness definition in Equation 2, we use the Markov inequality as follows

$$Adv_\epsilon^{\alpha,\beta}[f] = \mathbb{P}_{(G,X)\sim\mathcal{D}_{\mathcal{G},\mathcal{X}}}[(\tilde{G},\tilde{X})\in B_{\alpha,\beta}(G,X,\epsilon) \text{ s.t. } d_{\mathcal{Y}}(f(\tilde{G},\tilde{X}), f(G,X)) > \sigma]$$

$$\le \frac{1}{\sigma}\mathbb{E}_{\substack{(G,X)\sim\mathcal{D}_{\mathcal{G},\mathcal{X}},\\(G',X')\in B_{\alpha,\beta}((G,X),\epsilon)}}[\|f(A,X) - f(A,X')\|_\infty]$$

$$\le \frac{1}{\sigma}\left(\prod_{l=1}^{L}\|W^{(l)}\|_{op}\right)\epsilon\max_{u\in\mathcal{V}}|\hat{w}_u|_1 .$$

Thus, the classifier $f$ is $((d^{0,1}, \epsilon), (d_\infty, \gamma))$–robust robust with

$$\gamma = \prod_{i=1}^{L}\|W^{(i)}\|\epsilon\max_{u\in\mathcal{V}}\hat{w}_u/\sigma.$$

And we also have that the classifier $f$ is $((d^{0,1}, \epsilon), (d_1, \gamma))$–robust with

$$\gamma = \prod_{i=1}^{L}\|W^{(i)}\|\epsilon(\sum_{u\in\mathcal{V}}\hat{w}_u)/\sigma.$$

$\square$

**Theorem.** *Let $f : (\mathcal{G}, \mathcal{X}) \to \mathcal{Y}$ denote a graph function composed of $L$ GCN layers, where $W^{(i)}$ denotes the weight matrix of the $i$-th layer. Further, let $d^{1,0}$ be a graph distance. For structural attacks with a budget $\epsilon$, the function $f$ is $((d^{1,0}, \epsilon), (d_2, \gamma))$–robust with*

$$\gamma = \prod_{i=1}^{L}\|W^{(i)}\|\|X\|\epsilon(1 + L\prod_{i=1}^{L}\|W^{(i)}\|)/\sigma.$$

*Proof.* Following the same analogy, let's consider the case of structural perturbations. We only consider that the activation functions $(\sigma^{(\ell)})_{1 \le \ell \le L}$ are *nonexpensive*. In this setting, the graph shift operator (the normalized adjacency matrix in our work) is the one to be edited and the input features are shared between the two graphs. Let $\tilde{A}$ the clean adjacency and $h^{(i)}$ its hidden represented in the $i$-th layer and $\tilde{A}'$, $h'^{(i)}$, with the attacked/perturbed one.

Let's first consider the following

$$\|h'^{(l)}\| = \|\sigma^{(l)}(\tilde{A}_2 h'^{(l-1)} W^{(l)})\|$$
$$\le \|\tilde{A}_2\| \|W^{(l)}\| \|h'^{(l-1)}\|$$
$$\le \|W^{(l)}\| \|h'^{(l-1)}\|.$$

By recursion, we have: $\|h'^{(L)}\| \le \prod_{i=1}^{L} \|W^{(i)}\| \|X\|$.

From another side, we have the following

$$\|\sigma^{(l)}(\tilde{A} h_1^{(l)} W^{(l)}) - \sigma^{(l)}(\tilde{A}' h'_2^{(l)} W^{(l)})\|_2 \le \|\tilde{A} h^{(l)} W^{(l)} - \tilde{A}' h'_2^{(l)} W^{(l)}\|_2$$
$$\le \|W^{(l)}\| \|\tilde{A} h^{(l)} - \tilde{A}' h'^{(l)} + \tilde{A} h'^{(l)} - \tilde{A} h'^{(l)}\|_2$$
$$\le \|W^{(l)}\| \|\tilde{A}(h^{(l)} - h'^{(l)}) - h'^{(l)}(\tilde{A} - \tilde{A})\|_2$$
$$\le \|W^{(l)}\| (\|\tilde{A}\| \|(h^{(l)} - h'^{(l)})\| + \|h'^{(l)}\| \|(\tilde{A} - \tilde{A}')\|_2)$$
$$\le \|W^{(l)}\| \|(h^{(l)} - h'^{(l)})\| + \|W^{(l)}\| \|h'^{(l)}\| \epsilon.$$

By recursion, we get the following (By considering $\forall i > 1 : \|W^{(i)}\| \ge 1$)

$$\|\Phi(A_1, X) - \Phi(A_2, X)\|_2 \le \prod_{i=1}^{L} \|W^{(i)}\| \|X\| \epsilon + L(\prod_{i=1}^{L} \|W^{(i)}\|)^2 \|X\| \epsilon$$
$$\le \prod_{i=1}^{L} \|W^{(i)}\| \|X\| \epsilon (1 + L \prod_{i=1}^{L} \|W^{(i)}\|).$$

Similarly to the previous proof, we deduce that the classifier $f$ is $((d^{1,0}, \epsilon), (d_2, \gamma))$–robust where

$$\gamma = \prod_{i=1}^{L} \|W^{(i)}\| \|X\| \epsilon (1 + L \prod_{i=1}^{L} \|W^{(i)}\|)/\sigma.$$

$\square$

# D  PROOF OF THEOREM 4.3

**Theorem.** *Let $f : (\mathcal{G}, \mathcal{X}) \to \mathcal{Y}$ be composed of $L$ GIN-layers (with parameter $\zeta = 0$, that is usually denoted by $\epsilon$ in the literature) and $W^{(i)}$ denote the weight matrix of the $i$-th MLP layer. We consider the input node feature space to be bounded, i.e., $\|X\|_2 < B$ for some $B \in \mathbb{R}$, and graphs of maximum degree $\Delta_G$. For node feature-based attacks, with a budget $\epsilon$, the function $f$ is $((d^{0,1}, \epsilon), (d_\infty, \gamma))$–robust with*

$$\gamma = \prod_{l=1}^{L} \|W^{(l)}\|_\infty (B \, L \, \Delta_G + \epsilon)/\sigma.$$

*Proof.* Let's now consider a graph-function $f$ that is based on $L$ GIN-layers (with a parameter $\zeta = 0$, that is usually denoted by $\epsilon$ in the literature). The GIN message-passing propagation process can be written for a node $u$ as:

$$h_u^{(\ell+1)} = T^{(\ell+1)}((1+\zeta)h_u^{(\ell)} + \sum_{v \in \mathcal{N}(u)} h_v^{(\ell)})$$

where $T$ denotes a Neural Networks (a MLP) for example and $\zeta$ denotes the parameter of the GIN. We recall that $h^{(0)} = X \in \mathbb{R}^{n \times d}$ is set to the initial node features.

Let $X$ the original node features and $\tilde{X}$ the perturbed features. Let's consider a node $u \in V$, we denote by $h_u$ its representation in the clean graph and $h'_u$ its representation in the attacked graph , we note that we consider the activation functions to be *nonexpensive*. Since we only consider node feature based adversarial attacks, then by definition the original node $u$ and its corresponding corrupted node have the same neighborhood, we can therefore write

$$\|h_u^{(L)} - h'^{(L)}_u\| = \|T^{(\ell)}((1+\zeta)h_u^{(\ell-1)} + \sum_{v \in \mathcal{N}(u)} h_v^{(\ell-1)}) - T^{(\ell)}((1+\zeta)h'^{(\ell)}_u + \sum_{v \in \mathcal{N}(u)} h'^{(\ell)}_v)\|$$

$$\leq \|W^{(\ell)}\| \|(1+\zeta)(h_u^{(\ell-1)} - h'^{(\ell-1)}_u) + \sum_{v \in \mathcal{N}(u)} (h_v^{(\ell-1)} - h'^{(\ell-1)}_v)\|$$

We assume that the input feature space $\mathcal{H}_0$ is bounded, thus each hidden space $\mathcal{H}_i$ of the iterative process of message passing is bounded and let $B = \max_{\ell \leq L} B_\ell$ be its global maximum bound. From the previous result, we have

$$\|h_u^{(\ell)} - h'^{(\ell)}_u\| \leq \|W^{(\ell)}\| \|(1+\zeta)(h_u^{(\ell-1)} - h'^{(\ell-1)}_u) + B \times deg(u)\|$$

By iteration over the previous process, and by considering $\forall i > 1 : \|W^{(i)}\| \geq 1$, we can write

$$\|h_u^{(L)} - h'^{(L)}_u\| \leq B \times deg(u) \times \prod_{l=1}^{L} \|W^{(l)}\| \sum_{l=1}^{L} (1+\zeta)^l + \prod_{l=1}^{L} \|W^{(l)}\| (1+\zeta)^L \|h_u^{(0)} - h'^{(0)}_u\|$$

$$\leq B \times deg(u) \times \prod_{l=1}^{L} \|W^{(l)}\| \sum_{l=1}^{L} (1+\zeta)^l + \prod_{l=1}^{L} \|W^{(l)}\| (1+\zeta)^L \epsilon$$

$$\leq \prod_{l=1}^{L} \|W^{(l)}\| [B \times deg(u) \sum_{l=1}^{L} (1+\zeta)^l + (1+\zeta)^L \epsilon]$$

Since we consider the GIN-parameter $\zeta \approx 0$, we can deduce from the previous result that

$$\|h_u^{(\ell+1)} - h'^{(\ell+1)}_{u'}\| \leq \prod_{l=1}^{L} \|W^{(l)}\| [B \times L \times deg(u) + \epsilon]$$

Let's now consider the final output of the model which represents in the case of node classification the individual output of each node.

$$\|f(A, X) - f(A, X')\| = \| \begin{bmatrix} \vdots \\ h_u^{(L)} - h'^{(L)}_u \\ \vdots \end{bmatrix} \|$$

Based on the previous formulation, we have the following results

$$\|f(A, X) - f(A, X')\|_\infty = \prod_{l=1}^{L} \|W^{(l)}\|_\infty [B \times L \times \max_{u \in \mathcal{V}} deg(u) + \epsilon]$$

Similar to the previous proof of Theorem 4.1, we connect the Lipschitz constant of our considered model $f$ to the robustness definition in Equation 2 through Markov inequality and we get the following

$$\gamma = \prod_{l=1}^{L} \|W^{(l)}\|_\infty [B \times L \times \max_{u \in \mathcal{V}} deg(u) + \epsilon]/\sigma.$$

$\square$

# E  GENERALIZING TO ANY GRAPH NEURAL NETWORK

While this work focuses on GCNs, it can be easily extended to other GNN architecture. Given the iterative nature of GNNs, they can be viewed as a composition of multiple continuously differentiable functions, $f_i : \mathcal{H}_i \rightarrow \mathcal{H}_{i+1}$, where $\mathcal{H}_i$ represents the input space of the $i - th$ function. The Lipschitz constant, as shown in Zhao et al. (2021) is given by $C = \prod_i \|\nabla f_i\|$. The functions $\{f_i\}_i$ can be grouped into two categories : **(i)** The set $\mathcal{L}$ of functions $f_i$ that are linear with a corresponding parameter $W_i$. **(ii)** The set of functions that are not linear. We can hence decompose $C'$ according to the nature of the function as follows

$$C' = \Big( \prod_{f_i \notin \mathcal{L}} \|\nabla f_i\| \Big) \Big( \prod_{f_i \in \mathcal{L}} \|W_i\| \Big).$$

We can assume the input feature space $\mathcal{H}_0$ to be bounded; this a realistic assumption. Thus, each hidden space $\mathcal{H}_i$ is also bounded, and since $\{\nabla f_i\}_{f_i \notin \mathcal{L}}$ are continuous functions on bounded spaces, there exists an upper bound $C_i$ for $\|\nabla f_i\|$. Therefore, there exists $C > 0$, such that,

$$C' \leq C \prod_{f_i \in \mathcal{L}} \|W_i\|.$$

We return to the previous case by simply taking $\epsilon' = \epsilon C/C'$. Furthermore, following the assumption on the input feature space being bounded, it is possible to derive an upper bound on the GNN's robustness when subject to both structural and feature-based attacks simultaneously (See Appendix F).

# F  VULNERABILITY UPPER-BOUND WHEN DEALING WITH BOTH STRUCTURAL AND NODE FEATURES ATTACKS

In this section, we derive the upper-bound for a GCN's vulnerability $Adv_{\alpha,\beta,\epsilon}[f]$ when dealing with both structural and feature-based attacks simultaneously.

$$
\begin{aligned}
Adv_{\alpha,\beta,\epsilon}[f] &\leq \frac{1}{\sigma} \mathbb{E}_{\substack{(G_1,X_1)\sim\mathcal{D}_{\mathcal{G},\mathcal{X}}, \\ (G_2,X_2)\in B_{\alpha,\beta}((G_1,X_1),\epsilon)}} [\|f(A_1, X_1) - f(A_2, X_2)\|_2] \\
&\leq \frac{1}{\sigma} \mathbb{E}_{\substack{(G_1,X_1)\sim\mathcal{D}_{\mathcal{G},\mathcal{X}}, \\ (G_2,X_2)\in B_{\alpha,\beta}((G_1,X_1),\epsilon)}} [\|f(A_1, X_1) - f(A_1, X_2) + f(A_1, X_2) - f(A_2, X_2)\|_2] \\
&\leq \frac{1}{\sigma} \mathbb{E}_{\substack{(G_1,X_1)\sim\mathcal{D}_{\mathcal{G},\mathcal{X}}, \\ (G_2,X_2)\in B_{\alpha,\beta}((G_1,X_1),\epsilon)}} [\|f(A_1, X_1) - f(A_1, X_2)\|_2 + \|f(A_1, X_2) - f(A_2, X_2)\|_2] \\
&\leq \frac{1}{\sigma} \mathbb{E}_{\substack{(G_1,X_1)\sim\mathcal{D}_{\mathcal{G},\mathcal{X}}, \\ (G_2,X_2)\in B_{\alpha,\beta}((G_1,X_1),\epsilon)}} [\|f(A_1, X_1) - f(A_1, X_2)\|_2] + \\
&\qquad \mathbb{E}_{\substack{(G_1,X_1)\sim\mathcal{D}_{\mathcal{G},\mathcal{X}}, \\ (G_2,X_2)\in B_{\alpha,\beta}((G_1,X_1),\epsilon)}} [\|f(A_1, X_2) - f(A_2, X_2)\|_2].
\end{aligned}
$$

Table 3: Attacked classification accuracy ($\pm$ standard deviation) of the models on different benchmark node classification datasets after both structural attacks and node feature attacks application.

| Dataset | GCN | GCN-Jaccard | RGCN | GNN-SVD | GNNGuard | ParsevalR | GCORN |
|---|---|---|---|---|---|---|---|
| Cora | $76.7 \pm 1.2$ | $76.7 \pm 0.4$ | $78.1 \pm 0.6$ | $65.9 \pm 4.0$ | $61.0 \pm 0.4$ | $77.6 \pm 1.1$ | $\mathbf{79.8 \pm 0.7}$ |
| CiteSeer | $68.7 \pm 0.3$ | $71.5 \pm 0.3$ | $68.9 \pm 0.8$ | $68.3 \pm 0.5$ | $65.0 \pm 0.3$ | $70.9 \pm 0.6$ | $\mathbf{74.2 \pm 0.3}$ |
| PubMed | $47.0 \pm 0.4$ | $49.4 \pm 0.8$ | $51.2 \pm 0.4$ | $47.7 \pm 0.5$ | $40.3 \pm 0.3$ | $49.7 \pm 1.1$ | $\mathbf{53.9 \pm 0.8}$ |
| CoraML | $63.3 \pm 0.0$ | $32.2 \pm 1.5$ | $59.7 \pm 1.4$ | $53.8 \pm 0.8$ | $32.2 \pm 1.5$ | $65.4 \pm 0.2$ | $\mathbf{77.4 \pm 1.1}$ |

We already established an upper-bound in Appendix C for each term of the previous inequality. Therefore, we can have a new upper-bound for $Adv_{\alpha,\beta,\epsilon}[f]$

$$Adv_{\alpha,\beta,\epsilon}[f] \leq \frac{1}{\sigma}\left(\prod_{i=1}^{L}\|W^{(i)}\|\epsilon(\sum_{u\in\mathcal{V}}\hat{w_u})^2\right)\epsilon + \frac{1}{\sigma}\left(\prod_{i=1}^{L}\|W^{(i)}\|\|X\|\epsilon(1 + L\prod_{i=1}^{L}\|W^{(i)}\|)\right).$$

Following the assumption on the input feature space being bounded, i.e. $\exists B > 0, \forall X \quad \|X\| \leq B$, the inquality become

$$Adv_{\alpha,\beta,\epsilon}[f] \leq \frac{1}{\sigma}\left(\prod_{i=1}^{L}\|W^{(i)}\|\right)\left((\sum_{u\in\mathcal{V}}\hat{w_u})^2 + B(1 + L\prod_{i=1}^{L}\|W^{(i)}\|)\right)\epsilon.$$

Thus, the classifier $f$ is $d^{0,1}$-$(\epsilon,\gamma)$ robust where

$$\gamma = \left(\prod_{i=1}^{L}\|W^{(i)}\|\right)\left((\sum_{u\in\mathcal{V}}\hat{w_u})^2 + B(1 + L\prod_{i=1}^{L}\|W^{(i)}\|)\right)\epsilon/\sigma.$$

### F.1 EXPERIMENTAL VALIDATION

We assessed the efficacy of our proposed method against both Structural and Node-feature-based adversarial attacks simultaneously. This evaluation involved a combined evasion attack approach, wherein two attacks were integrated. Initially, we generated the perturbed adjacency matrix by employing a surrogate GCN model along with the "Mettack" (utilizing the 'Meta-Self' strategy) (Zügner & Günnemann, 2019). Subsequently, we perturbed the node features using a random attack strategy involving the injection of Gaussian noise $\mathcal{N}(0, \mathbf{I})$ into the features. This perturbation was controlled by a scaling parameter $\psi$, which we identified as effective in prior experiments. For the structural perturbations, we set the perturbation budget to $0.1E$, while for the node features, $\psi$ was set to $5.0$. We compared our method to the same considered defense benchmarks that were used in Section 6.

## G TIME AND COMPLEXITY ANALYSIS

Drawing from the original paper (Björck & Bowie, 1971), it has been established that the Bjorck orthonormalization method will always converge, provided that the condition $\|W^T W - I\|_2 < 1$ is met. In this section, we will begin by examining the connection between the selected order, the number of iterations and the resulting performance and then present an empirical time analysis of the proposed GCORN on various datasets.

### G.1 ON THE EFFECT OF ORDER/ITERATIONS

As per the projection equation 5, it appears that a higher order/iteration generally results in a more precise projection of the weight matrix. In our study, we gauge the accuracy of the approximation by assessing the model's performance in the absence of adversarial attacks, as well as under attack. Although our experiments demonstrate that employing the first order with a restricted number of

iterations can produce satisfactory outcomes, we believe that an hyper-parameter analysis should be conducted. Consequently, we evaluate the influence of modifying the order (while maintaining a constant number of iterations) on the resulting accuracy in Table 4.

Table 4: Performance of GCN and our proposed GCORN model, for different used approximation orders, on the Cora dataset.

|  | GCN | GCORN(1 ORD) | GCORN(2 ORD) | GCORN(3 ORD) |
|---|---|---|---|---|
| TRAINING TIME (IN S) | $2.8 \pm 0.01$ | $4.8 \pm 0.07$ | $8.7 \pm 0.07$ | $10.9 \pm 0.08$ |
| ACCURACY W/O ATTACK | $79.2 \pm 1.6$ | $78.8 \pm 1.3$ | $79.8 \pm 0.9$ | $80.8 \pm 1.1$ |
| ACCURACY W. ATTACK | $68.4 \pm 1.9$ | $77.1 \pm 2.1$ | $78.3 \pm 1.1$ | $78.6 \pm 0.4$ |

Table 4 shows both clean and attacked accuracy of the GCN and our proposed GCORN and the corresponding standard deviations for different orders. We observe that using higher order approximations generally leads to enhanced accuracy and robustness, albeit at the expense of increased running time. Therefore, the choice of order hyperparameter employed in the approximation must be carefully tuned for each application to find the best trade-off between robustness, accuracy and training time. In our experiments, we found that an approximation of order 1 yielded adequate and satisfactory results.

### G.2    ON TRAINING TIME

In line with the above, we also investigate the training time of our model compared to that of a standard GCN. To conduct this experiment, we held the order and number of iterations constant and utilized identical architectures and hyperparameters for both models (see Appendix L).

Table 5: Mean training time analysis (in s) of a our GCORN in comparison to the other benchmarks.

| DATASET | GCN | GCN-K | AIRGNN | RGCN | GCORN |
|---|---|---|---|---|---|
| CORA | 2.8 | 1.8 | 2.6 | 3.2 | 4.8 |
| CITESEER | 2.4 | 5.8 | 2.9 | 2.4 | 4.6 |
| PUBMED | 5.9 | 8.9 | 7.4 | 14.5 | 7.3 |
| CS | 6.1 | 12.1 | 12.4 | 13.8 | 15.5 |
| OGBN-ARXIV | 77.8 | 185.8 | 68.1 | 161.6 | 78.4 |

The mean training time are presented in Table 5, indicating a trade-off between the improved robustness provided by the GCORN model and its slightly longer training time compared to the GCN model and other available methods.

## H    MORE DETAILS ABOUT THE ESTIMATION OF OUR ROBUSTNESS MEASURE

We begin by providing the pseudocode of the algorithm used to estimate our robustness measure.

---
**Algorithm 1:** Estimation of $Adv_\epsilon^{\alpha,\beta}[f]$.

---
**Inputs:** Sphere Radius : $\epsilon > 0$, Number of Samples $L_{max}$, Number of Input Graphs $|\mathcal{D}|$;
Initialize $Adv = 0$;
**foreach** $[G_i, X_i] \in \mathcal{D}$ **do**
    Initialize $Adv_i = 0$;
    **foreach** $l = 1, \ldots, L_{max}$ **do**
        1. Sample a distance $r \in [0, \epsilon]$ from the prior distribution $p_\epsilon$ (see Appendix I);
        2. Uniformly sample $Z_l \in \mathbb{R}^{n \times K}$ from $S_r$ (see Appendix H);
        3. Choose $\tilde{X}_l = X_i + Z_l$;
        4. Update
            $Adv_i \leftarrow Adv_i + \mathbf{1}\{d_{\mathcal{Y}}(f(\tilde{G}_l, \tilde{X}_l), f(G, X)) > \sigma\}$;
    **end foreach**
    $Adv_i = Adv_i / L_{max}$; $Adv = Adv + Adv_i$;
**end foreach**
Return $Adv / |\mathcal{D}|$ ;

---

We now detail how to uniformly sample from the set $S_r$ defined by,

$$S_r = \{Z \in \mathbb{R}^{n \times D} \mid \max_{1 \le i \le n} \|Z_i\|_p = r\},$$

where $r \in [0, \epsilon]$. The set $S_r$ can also be defined in the following way,

$$Z \in S_r \Leftrightarrow \max_{1 \le i \le n} \|Z_i\|_p = r$$

$$\Leftrightarrow \begin{cases} \exists i_0 \in \{1, \dots, n\} & \|Z_{i_0}\|_p = r, \\ \forall i \in \{1, \dots, n\} & \|Z_i\|_p \le r. \end{cases}$$

Since the position of $i_0$ within $\{1, \dots, n\}$ do not matter in the previous equivalence, we can uniformly sample $i_0 \in \{1, \dots, n\}$, such that $r_{i_0} = \|Z_{i_0}\|_p = r$ and that the other index $i \in \{1, \dots, n\} \setminus \{i_0\}$ should satisfy $r_i = \|Z_i\|_p \le r$, i.e., $Z_i \in \mathbb{B}^K(r) = \{x \in \mathbb{R}^K \mid \|x\|_p \le r\}$. We will use another time *Stratified Sampling* where we start by sampling $\{r_i\}_{i \ne i_0}$ within $[0, r]$. Using the Lemma 5.1, we can directly sample $\{r_i\}_{i \ne i_0}$ from the probability distribution

$$\forall i \in \{1, \dots, n\} \setminus \{i_0\}, \quad p(r_i) = K \frac{1}{r} \left(\frac{r_i}{r}\right)^{K-1}.$$

Once we know the values of $\{r_i\}_i$, the problem boils down to sampling $n$ vectors $\{Z_i\}_{1 \le i \le n}$ from $\mathbb{R}^K$, such that

$$\forall i \in \{1, \dots, n\}, \quad \|Z_i\|_p = r_i. \tag{16}$$

For that, we have to consider the two cases $p < \infty$ and $p = \infty$

## H.1 CASE WHERE $p < \infty$

In this case, Equation 16 can be written as follows

$$\forall i \in \{1, \dots, n\}, \quad \|Z_i\|_p = r_i \Leftrightarrow \forall i \in \{1, \dots, n\}, \quad \sum_{j=1}^{K} |Z_{i,j}|^p = r_i^p,$$

$$\Leftrightarrow \forall i \in \{1, \dots, n\}, \quad \sum_{j=1}^{K} \left(\frac{|Z_{i,j}|}{r_i}\right)^p = 1. \tag{17}$$

To satisfy Equation (17), we randomly uniformly partition the $[0, 1]$ into $K$ parts. To do so, for each $i \in \{1, \dots, n\}$, we randomly sample $D - 1$ element from uniform distribution $\mathcal{U}[0, 1]$; Let's denote $[p_1^{(i)}, \dots, p_{K-1}^{(i)}]$ the sorted list of sampled elements. We directly choose

$$\mathcal{O}_{i,j} = \begin{cases} 1 - p_{j-1}^{(i)} & \text{if } j = K, \\ p_j^{(i)} & \text{if } j = 0, \\ p_j^{(i)} - p_{j-1}^{(i)} & \text{otherwise}. \end{cases}$$

The matrix $\mathcal{O} = (\mathcal{O}_{i,j})_{1 \le i \le n, 1 \le j \le D}$ satisfy

$$\forall i \in \{1, \dots, n\}, \quad \sum_{j=1}^{K} \mathcal{O}_{i,j} = 1.$$

Figure 2 gives insight into the previously introduced concept of randomly partitioning in the case of the set $[0, 1]$ into $K = 4$.

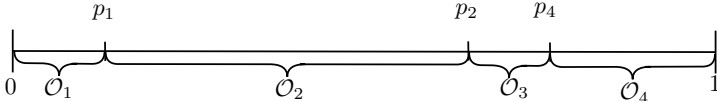

Figure 2: A toy example on how to randomly partition the set $[0, 1]$ into $K = 4$ parts such as the sum of the parts lengths is 1. We first uniformly sample 3 elements from $[0, 1]$ and reorder them $[p_1, p_2, p_3]$. And subsequently, we consider $\mathcal{O}_1 = [0, p_1]$, $\mathcal{O}_2 = [p_4, p_3]$, $\mathcal{O}_3 = [p_4, p_3]$, $\mathcal{O}_4 = [1, p_4]$

Based on the previous elements, we can choose $Z$ such that

$$\forall i, j, \quad \left(\frac{|Z_{i,j}|}{r_i}\right)^p = \mathcal{O}_{i,j} \Leftrightarrow \forall i, j, \quad |Z_{i,j}| = r_i \times \mathcal{O}_{i,j}^{1/p}. \tag{18}$$

Equation 18 is invariant with respect to the sign of $Z_{i,j}$, hence, we use the following

$$\forall i, j, \quad Z_{i,j} = u_{i,j} \times r_i \times \mathcal{O}_{i,j}^{1/p},$$

where $u_{i,j}$ is sampled from the discrete uniform distribution $\mathcal{U}_{\{-1,1\}}$.

We recall that for $p = 1$, the distance $d^{0,1}([G, X], [\tilde{G}, \tilde{X}]) = \max_{i \in \{1,\ldots,n\}} \|X_i - \tilde{X}_i\|_p$ defined in Equation 7 matches with the infinity matrix distance $d_\infty : (A, B) \mapsto \sum_{X \neq 0} \frac{\|AX - BX\|_\infty}{\|X\|_\infty}$ as shown in Nagisa (2022). Therefore, from the Theorem 4.1, the upper-bound of the expected vulnerability $Adv_\epsilon^{\alpha,\beta}[f]$ is $\gamma = \prod_{i=1}^L \|W^{(i)}\|_\infty \epsilon \hat{w}_G / \sigma$.

## H.2 Case where $p = \infty$

In this case, Equation 16 can be written as follows

$$\forall i \in \{1, \ldots, n\}, \quad \|Z_i\|_\infty = r_i \Leftrightarrow \forall i \in \{1, \ldots, n\}, \quad \max_{1 \leq j \leq K} |Z_{i,j}| = r_i$$

$$\Leftrightarrow \forall i \in \{1, \ldots, n\}, \quad \begin{cases} \exists j_0 \in \{1, \ldots, K\} & |Z_{i,j_0}| = r_i, \\ \forall j \in \{1, \ldots, K\} & |Z_{i,j_0}| \leq r_i. \end{cases} \tag{19}$$

For each $i \in \{1, \ldots, n\}$, we randomly select $j_0 \in \{1, \ldots, K\}$ to satisfy $|Z_{i,j_0}| = r_i$. For the other $j \neq j_0$, we uniformly sample the value of $|Z_{i,j}|$ from $\mathcal{U}_{[0,r_i]}$. These equations are invariant with respect to the sign of $Z_{i,j}$, therefore, we choose,

$$\forall i, j, \quad Z_{i,j} = u_{i,j} \times |Z_{i,j}|,$$

where $u_{i,j}$ is sampled from the discrete uniform distribution $\mathcal{U}_{\{-1,1\}}$.

## I Proof of Lemma 5.1

**Lemma.** *Let $\mathbb{R}^K$ be the real finite-dimensional space and $\epsilon$ a positive real number. If $R^{(p)}$ is the random variable indicating the maximum of the $L_p$ norms values inside the ball of radius $\epsilon$, i.e., $\mathcal{B}_\epsilon = \{Z \in \mathbb{R}^{n \times K} : \max_{i \in \{1,\ldots,n\}} \|Z_i\|_p \leq \epsilon\}$. Then, for every $p > 0$, the density distribution of $R^{(p)}$ does not depends on $p$ and is defined as follows,*

$$p_\epsilon(r) = K \frac{1}{\epsilon} \left(\frac{r}{\epsilon}\right)^{K-1} \mathbf{1}\{0 \leq r \leq \epsilon\}.$$

*Proof.* Using the following equivalence

$$\exists Z \in \mathbb{R}^{n,K} \max_{1 \leq i \leq n} \|Z_i\|_p = r \Leftrightarrow \exists T \in \mathbb{R}^K, \ \|T\|_p = r.$$

We deduce that the density $p_\epsilon$ do not depend on $n$, and therefore we can set $n = 1$ for this proof. Therefore,

$$\mathcal{B}_\epsilon = \{Z \in \mathbb{R}^K | \|Z\|_p \leq \epsilon\}.$$

We start by calculating the volume of the $K-$sphere of radius $r$ for any real finite-dimensional space $\mathbb{R}^K$ where $K > 2$ is an integer.

$$\mathbb{S}^K(r) = \{x \in \mathbb{R}^K | \|x\|_p = r\}.$$

Let's denote the volume of $\mathbb{S}^K(r)$ by $\mathcal{V}^K(r)$.

Using *Fubini Theorem*, we have

$$
\begin{aligned}
\mathcal{V}^K(r) &= \int_{-r}^{r} \mathcal{V}^{K-1}\left(\left(r^p - |x|^p\right)^{1/p}\right) dx \\
&= r \int_{-1}^{1} \mathcal{V}^{K-1}\left(r\left(1 - |x|^p\right)^{1/p}\right) dx \\
&= r \int_{-1}^{1} r^{K-1}\mathcal{V}^{K-1}\left(\left(1 - |x|^p\right)^{1/p}\right) dx \\
&= r^K \int_{-1}^{1} \mathcal{V}^{K-1}\left(\left(1 - |x|^p\right)^{1/p}\right) dx \\
&= r^K \mathcal{V}^K(1).
\end{aligned}
\tag{20}
$$

Thus, the surface $L^p$ ball in $\mathbb{R}^K$ of radius $r$ is

$$\mathcal{S}^K(r) = \frac{d\mathcal{V}^K(r)}{dr} \tag{21}$$

$$= Kr^{K-1}\mathcal{V}^K(1). \tag{22}$$

Consequently the density distribution of $\mathbf{R}^{(p)}$ could be written as

$$
\begin{aligned}
p_\epsilon(r) &= \frac{\mathcal{S}_K^p(r)}{\mathcal{V}_K^p(\epsilon)}\mathbf{1}\{0 \leq r \leq \epsilon\} \\
&= \frac{1}{\mathcal{V}_K^p(\epsilon)}\frac{d\mathcal{V}_K^p(r)}{dr}\mathbf{1}\{0 \leq r \leq \epsilon\} \\
&= a\frac{1}{\epsilon}\left(\frac{r}{\epsilon}\right)^{K-1}\mathbf{1}\{0 \leq r \leq \epsilon\}.
\end{aligned}
\tag{23}
$$

We note that the previous quantity doesn't depend on $p$. $\qquad\square$

## J  ADDITIONAL RESULTS

### J.1  NODE CLASSIFICATION

To further understand the relationship between robustness and input sensitivity, from which we derived the upper bound for GNNs and specifically GCNs, we empirically validate the trade-off between robustness and input perturbation. We hence compare the difference in the output of each model when subjected to random perturbations with varying attack budgets. The results of this study are presented in Figure 3. The figure displays the results for both Cora and CiteSeer datasets in both absolute (left figure) and log-scale (right figure) format. The results demonstrate that the GCN is highly sensitive to perturbations in comparison to the proposed approach in terms of the difference in output. These findings provide validation for the stability of GCORN against input perturbations, and by extension, adversarial attacks.

We additionally considered two defense techniques benchmarks: **(1)** ElasticGNN (Liu et al., 2021b) which proposes a novel and general message passing scheme into GNNs to enhance the local smoothness adaptivity of GNNs via $\ell_1$-based graph smoothing and **(2)** EvenNet (Lei et al., 2022) which proposes a spectral GNN corresponding to even-polynomial graph filter with the main idea being that ignoring odd-hop neighbors improves the underlying robustness. For this additional evaluation, we consider similar attacks techniques as the one from Table 1.

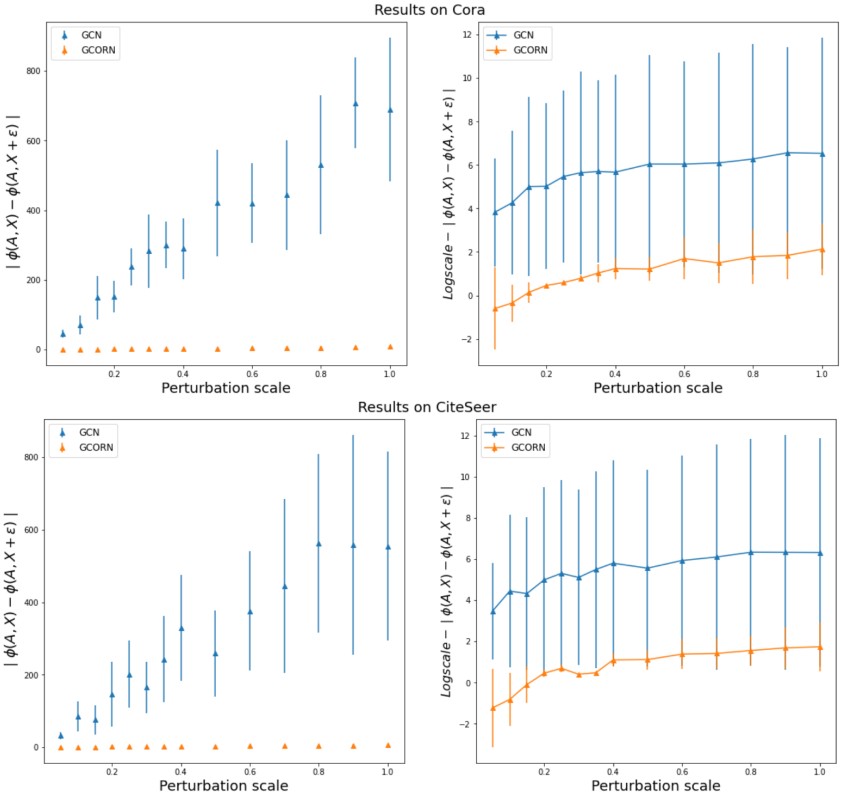

Figure 3: Difference (in average and standard deviation) in output for the GCN and our GCORN when subject to random perturbations with different attack budgets for both (a) Cora and (b) CiteSeer. The right-hand side plots are on the log-scale.

Table 6: Attacked classification accuracy ($\pm$ standard deviation) of the models on different benchmark node classification dataset after the attack application.

| Attack | Dataset | ElasticGNN | EvenNet | GCORN |
|---|---|---|---|---|
| Random ($\psi = 0.5$) | Cora | $72.5 \pm 2.4$ | $69.7 \pm 1.8$ | $77.1 \pm 1.8$ |
| | CiteSeer | $58.9 \pm 2.3$ | $48.6 \pm 1.6$ | $67.8 \pm 1.4$ |
| | PubMed | $70.5 \pm 0.6$ | $68.8 \pm 1.6$ | $73.1 \pm 1.1$ |
| Random ($\psi = 0.5$) | Cora | $54.4 \pm 2.4$ | $48.9 \pm 2.1$ | $57.6 \pm 1.9$ |
| | CiteSeer | $43.8 \pm 2.0$ | $39.7 \pm 4.2$ | $57.3 \pm 1.7$ |
| | PubMed | $56.8 \pm 2.1$ | $58.5 \pm 3.2$ | $65.8 \pm 1.4$ |
| PGD | Cora | $64.7 \pm 1.6$ | $57.6 \pm 3.6$ | $71.1 \pm 1.4$ |
| | CiteSeer | $50.8 \pm 3.2$ | $37.1 \pm 3.3$ | $65.6 \pm 1.4$ |
| | PubMed | $64.6 \pm 1.5$ | $61.0 \pm 4.1$ | $72.3 \pm 1.3$ |

## J.2 GRAPH CLASSIFICATION

In this section, we present the result of our *GCORN* on the graph classification task using both empirical and our proposed probabilistic evaluation. Details about the experimental setting for this task are provided in Appendix L. Note that for the random attack, we used $\psi = 0.5$ and for the gradient-based, we used a budget $\delta = 0.2$. Table 7 reports the average accuracy and the corresponding standard deviation for both clean and attacked accuracy.

Table 7: Classification accuracy ($\pm$ standard deviation) of the models on different benchmark graph classification dataset before (clean) and after the attack. The higher the accuracy (in %) the better.

| Attack | PROTEINS | | D&D | | NCI1 | |
|---|---|---|---|---|---|---|
| | **GCN** | **GCORN** | **GCN** | **GCORN** | **GCN** | **GCORN** |
| Clean | $73.4 \pm 2.8$ | $74.1 \pm 1.9$ | $75.8 \pm 3.6$ | $76.4 \pm 4.1$ | $75.7 \pm 2.2$ | $74.8 \pm 1.7$ |
| Random | $66.7 \pm 2.5$ | $71.7 \pm 2.8$ | $70.9 \pm 3.2$ | $75.1 \pm 4.8$ | $67.1 \pm 2.6$ | $71.8 \pm 2.1$ |
| PGD | $56.7 \pm 2.8$ | $65.4 \pm 3.1$ | $61.8 \pm 4.1$ | $68.6 \pm 4.3$ | $54.9 \pm 2.9$ | $62.4 \pm 2.1$ |

## K    EXPERIMENTAL RESULTS ON GIN

To showcase our method's versatility beyond the considered GCN framework and confirm the GIN-related results outlined in Theorem 4.3, we conducted similar experiments to those presented in Table 1. However, this time, we employed GIN as the message-passing scheme. In this perspective, we used the same previously considered node features attacks, notably: **(i)** The baseline random attack injecting Gaussian noise $\mathcal{N}(0, \mathbf{I})$ to the features with a scaling parameter $\psi$ controlling the attack budget; **(ii)** The white-box Proximal Gradient Descent (Xu et al., 2019a), which is a gradient-based approach to the adversarial optimization task with a chosen attack budget of 15%.

Table 8: Attacked classification accuracy ($\pm$ standard deviation) of the models on different benchmark node classification dataset using Graph Isomorphism Network after the attack application.

| | Model | Cora | CiteSeer | PubMed |
|---|---|---|---|---|
| Random ($\psi = 0.5$) | GIN | $67.8 \pm 1.3$ | $50.1 \pm 0.3$ | $72.8 \pm 0.4$ |
| | GIORN | $\mathbf{71.8 \pm 0.8}$ | $\mathbf{60.8 \pm 0.5}$ | $\mathbf{74.3 \pm 0.7}$ |
| Random ($\psi = 1.0$) | GIN | $54.5 \pm 0.9$ | $45.1 \pm 0.4$ | $67.3 \pm 0.6$ |
| | GIORN | $\mathbf{66.7 \pm 1.2}$ | $\mathbf{54.4 \pm 0.8}$ | $\mathbf{68.9 \pm 1.1}$ |
| PGD | GIN | $61.4 \pm 1.4$ | $44.1 \pm 0.5$ | $70.6 \pm 0.4$ |
| | GIORN | $\mathbf{69.5 \pm 0.6}$ | $\mathbf{58.3 \pm 0.6}$ | $\mathbf{73.1 \pm 1.6}$ |

## L    DATASETS AND IMPLEMENTATION DETAILS

For all the used models, the same number of layers, hyperparameters, and activation functions were used. The models were trained using the cross-entropy loss function with the Adam optimizer, the number of epochs and learning rate were kept similar for the different approaches across all experiments.

### L.1    NODE CLASSIFICATION

Characteristics and information about the datasets utilized in the node classification part of the study are presented in Table 10. As outlined in the main paper, we conduct experiments on a set of citation networks, including Cora, CiteSeer, and PubMed (Sen et al., 2008), as well as the Amazon Co-author network of authors from the Computer Science (CS) domain (Shchur et al., 2018). For Cora, CiteSeer, and PubMed, we adhere to the train/valid/test splits provided by the datasets. For the CS dataset, we adopt the same methodology as used in Yang et al. (2016), by randomly selecting 20 nodes from each class to form the training set and 500/1000 nodes from the remaining for the validation and test sets.

In all of the experiments, the models employed a 2-layer convolutional architecture (consisting of two iterations of message passing and updating) stacked with a Multi-Layer Perception (MLP) as a readout. The intent was to compare the models in an iso-architectural setting, to ensure a fair evaluation of their robustness. All experiments were conducted using the Adam optimizer (Kingma & Ba, 2015) and the same hyperparameters, including a learning rate of 1e-2, 300 epochs, and a hidden feature dimension of 16. For the OGB dataset, we used 512 as a hidden feature dimension. To account for the impact of random initialization, each experiment was repeated 10 times, and the mean and standard deviation of the results were reported. We finally note that for the AIRGNN, we

Table 9: Statistics of the node classification datasets used in our experiments.

| DATASET | #FEATURES | #NODES | #EDGES | #CLASSES |
|---|---|---|---|---|
| CORA | 1433 | 2708 | 5208 | 7 |
| CITESEER | 3703 | 3327 | 4552 | 6 |
| PUBMED | 500 | 19717 | 44338 | 3 |
| CS | 6805 | 18333 | 81894 | 15 |
| OGBN-ARXIV | 128 | 31971 | 71669 | 40 |

Table 10: Statistics of the graph classification datasets used in our experiments.

| DATASET | #GRAPHS | #NODES | #EDGES | #CLASSES |
|---|---|---|---|---|
| DD | 1178 | 284.32 | 715.66 | 2 |
| NCI1 | 4110 | 29.87 | 32.30 | 2 |
| PROTEINS | 1113 | 39.06 | 72.82 | 2 |

set $K = 2$ so as to have the same number of propagation as the other benchmarks. For our proposed GCORN, we tuned the iteration numbers for the different datasets and this was also done for the Parseval Regularization (ParselR) to find the best regularization parameter.

## L.2 GRAPH CLASSIFICATION

We additionally emperically evaluated our proposed method GCORN using both probabilistic evaluation and the classical experimental evaluation. We used benchmark datasets derived from bioinformatics and chemoinformatics (PROTEINS, NCI1, D&D) (Morris et al., 2020). The framework proposed by Errica et al. (2020) was used to evaluate the performance of the models on this task. We therefore performed a 10-fold cross-validation using the same folds as provided by the paper to obtain an estimate of the generalization performance of each method.

Similar to node classification, we employed a 2-layer convolutional architecture (consisting of two iterations of message passing and updating) stacked with a Multi-Layer Perception (MLP) as a readout using the Adam Optimizer.

## L.3 IMPLEMENTATION DETAILS

Our implementation is available in the supplementary materials (and will be publicly available afterwards). It is built using the open-source library *PyTorch Geometric* (PyG) under the MIT license (Fey & Lenssen, 2019). We leveraged the publicly available implementation of the different benchmarks from their available repositories : From GCN-k [1], for AIRGNN [2], for GNNGuard [3] and for RGCN we used the implementation from the DeepRobust package. Note that we additionally utilized the PyTorch DeepRobust package[4] to implement the adversarial attacks used in this study. The experiments have been run on both a NVIDIA A100 GPU and a RTX A6000 GPU.

## L.4 IMPLEMENTATION DETAILS OF OUR EMPIRICAL ROBUSTNESS ESTIMATION

We considered the input distance defined in Equation (7) with $p = 2$, we fixed the radius $\epsilon = 10$ and the number of sampling per graph at $L_{max} = 100$. For each graph $G$, the output distance $d_{\mathcal{Y}}$ has been re-scaled to the interval $[0, 1]$ by normalization using a factor of $2\sqrt{N_G}$ where $N_G$ is the number of nodes in the graph $G$ (See Appendix M).

---

[1] https://github.com/ChangminWu/RobustGCN

[2] https://github.com/lxiaorui/AirGNN

[3] https://github.com/mims-harvard/GNNGuard

[4] https://github.com/DSE-MSU/DeepRobust

# M    RESULTS OF THE EMPIRICAL ESTIMATION OF OUR ROBUSTNESS MEASURE

## M.1    NORMALIZATION OF THE OUTPUT DISTANCE $\mathbf{d}_{\mathcal{Y}}$

The distance $\mathbf{d}_{\mathcal{Y}}$ is defined as follows,

$$d_{\mathcal{Y}}(f(\tilde{G}, \tilde{X}), f(G, X)) = \|f(\tilde{G}, \tilde{X}) - f(G, X)\|_2.$$

Let's put $Q = (Q_{i,j})_{\substack{0 \leq i \leq N_G \\ 0 \leq i \leq C}} = f(\tilde{G}, \tilde{X}) - f(G, X)$, where $N_G$ is the number of nodes in the graph and $C$ the number of classes, i.e. output dimension. Since, the last layer of $f$ is a Sigmoid function, all the element of the matrix $f(\tilde{G}, \tilde{X})$ and $f(G, X)$ are bounded between 0 and 1, thus, $\forall i, j, \ |Q_{i,j}| \leq \|f(\tilde{G}, \tilde{X})\|_\infty + \|f(G, X)\|_\infty \leq 2$, i.e. $\|Q\|_\infty \leq 2$. Therefore, we have

$$
\begin{aligned}
d_{\mathcal{Y}}(f(\tilde{G}, \tilde{X}), f(G, X)) &= \|S\|_2 \\
&= \sup_{x \neq 0} \frac{\|Sx\|_2}{\|x\|_2}, \\
&= \sup_{x \neq 0} \frac{\left(\sum_{i=1}^{N_G} \left(\sum_{j=1}^{C} Q_{i,j} x_j\right)^2\right)^{1/2}}{\|x\|_2} \\
&\leq \|Q\|_\infty \sup_{x \neq 0} \frac{\left(\sum_{i=1}^{N_G} \left(\sum_{j=1}^{C} x_j\right)^2\right)^{1/2}}{\|x\|_2} \\
&= \|Q\|_\infty \sqrt{N_G} \sup_{x \neq 0} \frac{\left(\sum_{j=1}^{C} x_j\right)^2}{\|x\|_2} \\
&= \|Q\|_\infty \sqrt{N_G} \sup_{x \neq 0} \frac{\|x\|_2}{\|x\|_2} \\
&= \|Q\|_\infty \sqrt{N_G} \leq 2\sqrt{N_G}.
\end{aligned}
$$

Thus,

$$0 \leq \frac{1}{2\sqrt{N_G}} d_{\mathcal{Y}}(f(\tilde{G}, \tilde{X}), f(G, X)) \leq 1.$$

## M.2    THE EFFECT OF SAMPLING ON THE EMPIRICAL ESTIMATION OF $Adv_\epsilon^{\alpha,\beta}[f]$

The provided robustness evaluation is based on an estimation related to a sampling strategy. Consequently, the quality of our evaluation is dependent on the number of generated samples. In what follows we aim to shed light on the effect of the chosen number of samples on our estimation.

Previous work (Niederreiter, 1992; Tezuka, 2012) has proven the intricate link between the discrepancy of samples and the estimation errors. Discrepancy, particularly in the context of quasi-Monte Carlo methods, assesses how effectively a set of points covers a given space. A high discrepancy signifies the presence of volumes with varying point densities—some with high density and others with low density. Conversely, a low discrepancy indicates a more uniform distribution of points, resulting in smaller estimation errors for $Adv_\epsilon^{\alpha,\beta}[f]$.

We use this concept to understand the possible variation of our estimation within our considered input graph's neighborhood. In this perspective, we measure the discrepancy by computing the probability to find a ball of radius $r \ll \epsilon$, included in $\mathcal{B}_\epsilon = \left\{Z \in \mathbb{R}^K | \|Z\|_p \leq \epsilon\right\}$, which does not contain any of the $L_{max}$ samples $\{Z_1, \ldots, Z_{L_{max}}\}$. We denote by $\mathbb{B}_r$ the set of balls of radius $r$, i.e. we have:

$$B \in \mathbb{B}_r \Leftrightarrow \exists T \in \mathbb{R}^K, \ B = \left\{Z \in \mathbb{R}^K | \|Z - T\|_p \leq \epsilon\right\}.$$

For any $B \in \mathbb{B}_r$ such that $B \subset \mathcal{B}_\epsilon$, the probability that none of the $L_{max}$ samples belongs to $B$ is

$$\mathbb{P}(\forall i \in \{1, \ldots, L_{max}\}, \ Z_i \notin B) = \prod_{i=1}^{L_{max}} \mathbb{P}(Z_i \notin B)$$
$$= \mathbb{P}(Z_1 \notin B)^{L_{max}}$$
$$= (1 - \mathbb{P}(Z_1 \in B))^{L_{max}}$$
$$= \left(1 - \frac{\mathcal{V}^K(r)}{\mathcal{V}^K(\epsilon)}\right)^{L_{max}}$$
$$= \left(1 - \left(\frac{r}{\epsilon}\right)^K\right)^{L_{max}}.$$

To reach a low discrepancy, we need to reduce the aforementioned probability, i.e. upper-bounding the probability $\mathbb{P}(\forall i \in \{1, \ldots, L_{max}\}, \ Z_i \notin B)$ with a small $\alpha$ within $]0, 1]$, e.g. $\alpha = 0.05$.

$$\mathbb{P}(\forall i \in \{1, \ldots, L_{max}\}, \ Z_i \notin B) \leq \alpha \Leftrightarrow \left(1 - \left(\frac{r}{\epsilon}\right)^K\right)^{L_{max}} \leq \alpha \tag{24}$$

$$\Leftrightarrow L_{max} log \left(1 - \left(\frac{r}{\epsilon}\right)^K\right) \leq log(\alpha) \tag{25}$$

$$\Leftrightarrow L_{max} \geq \frac{log(\alpha)}{log\left(1 - \left(\frac{r}{\epsilon}\right)^K\right)}. \tag{26}$$

Therefore, to reach a low discrepancy and hence reduce the estimation error, Equation 26 provides a theoretical guarantee for the minimum required number of samples. The given lower-bound is an increasing function in respect to $\epsilon$, thus, we may need fewer samples if we consider a smaller input graph's neighborhood.

We also empirically investigate how the robustness measure $Adv_\epsilon^{\alpha,\beta}[f]$ varies due to sampling. To do so, for a fixed number of samples, we empirically estimate $Adv_\epsilon^{\alpha,\beta}[f]$ 10 different times using the Algorithm 1. We report the mean and the standard deviation of $\hat{Adv}_\epsilon^{\alpha,\beta}[f]$ for each fixed number of samples. As noticed for the dataset Cora, we have a consistent estimation of $Adv_\epsilon^{\alpha,\beta}[f]$ with different number of samples. Moreover, the variance becomes almost constant with only 40 samples. Thus, with only a few number of samples, we can statistically significant estimation of $Adv_\epsilon^{\alpha,\beta}[f]$.

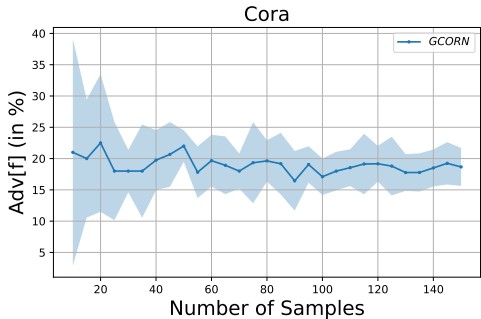

Figure 4: The Values of $Adv_\epsilon^{\alpha,\beta}[f]$ for the dataset Cora. The dotted line and the shaded region represent respectively the mean value and the standard deviation of $Adv_\epsilon^{\alpha,\beta}[f]$.

### M.3 EMPIRICAL VALIDATION OF THE TIGHTNESS OF THE COMPUTED THEORETICAL UPPER-BOUND

In this section, we study the tightness of the theoretical upper-bound $\gamma = \prod_{i=1}^{L} \|W^{(i)}\|_\infty \epsilon \hat{w}_G / \sigma$ of Theorem 4.1. For that, we use our empirical estimation of $Adv_\epsilon^{\alpha,\beta}[f]$, i.e. using Algorithm 1.

We calculated both its theoretical upper-bound and the estimated robustness evaluation for different values of $\sigma$. The results are provided in Figure 5. As shown in Appendix M.1, the maximum

possible value of $\sigma$ is $2\sqrt{N_G}$, thus, we plot the values of $Adv_\epsilon^{\alpha,\beta}[f]$ and the upper-bound $\gamma$ as a function of $\sigma/(2\sqrt{N_G})$. Notably, these experimental findings affirm the tightness of the theoretically computed bound. This tightness explains our considered link between reducing the upper-bound and the robustness enhancement as experimentally seen in Table 1 and 2.

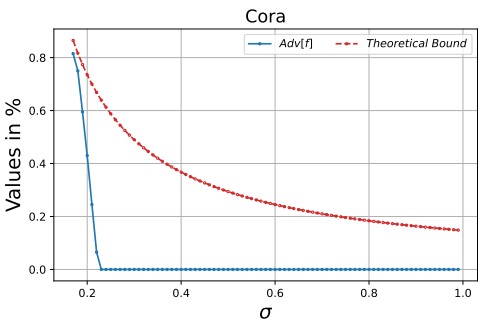

Figure 5: The Values of $Adv_\epsilon^{\alpha,\beta}[f]$ and the theoretical upper-bound $\gamma$ (c.f. Theorem 4.1) for the dataset Cora using different values of $\sigma$. For this experiment, we used the values $\epsilon = 0.1$ and $L_{max} = 100$.

### M.4 ADDITIONAL RESULTS OF THE EMPIRICAL ESTIMATION FOR THE NODES CLASSIFICATION TASK

In Figure 6, we report the values of $Adv_\epsilon^{\alpha,\beta}[f]$ of our GCORN and the baselines for the datasets CiteSeer, PubMed and CS. We used the same setting previously described in Section 5. For the dataset CiteSeer and PubMed, we notice that *GCORN* method yielded the lowest Adversarial Robustness value, indicating that *GCORN* exhibits greater robustness against adversarial examples.

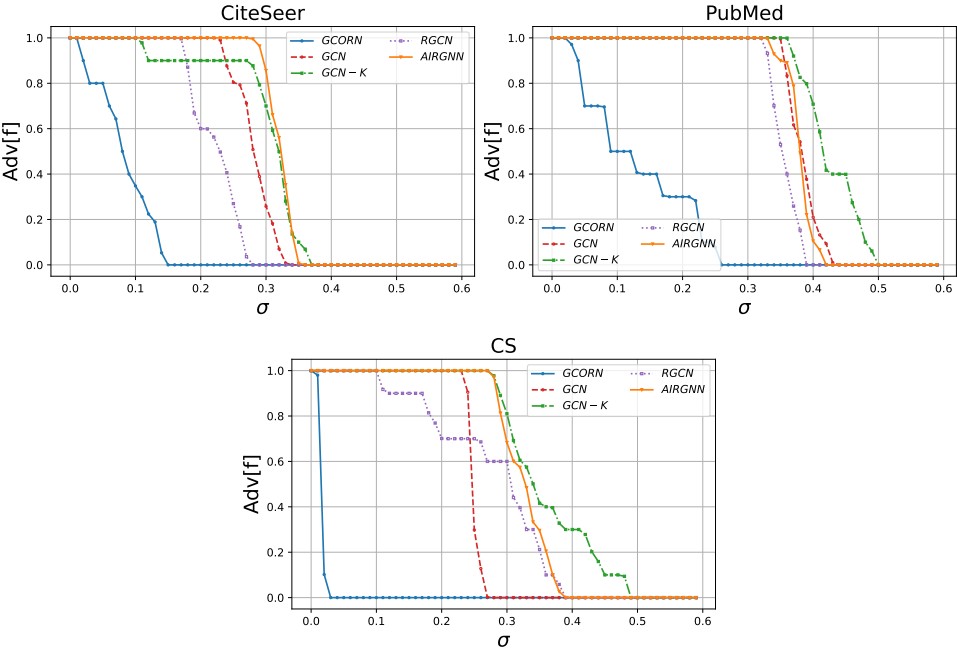

Figure 6: The Values of $Adv_\epsilon^{\alpha,\beta}[f]$ for the dataset CiteSeer, PubMed and CS.

## M.5 ADDITIONAL RESULTS OF THE EMPIRICAL ESTIMATION FOR THE GRAPHS CLASSIFICATION TASK

In Figures 7, we report the values of $Adv_\epsilon^{\alpha,\beta}[f]$ for the datasets DD, NCI and Proteins. We observe that GCORN and the two other baselines exhibit nearly identical values of Ad $Adv_\epsilon^{\alpha,\beta}[f]$.

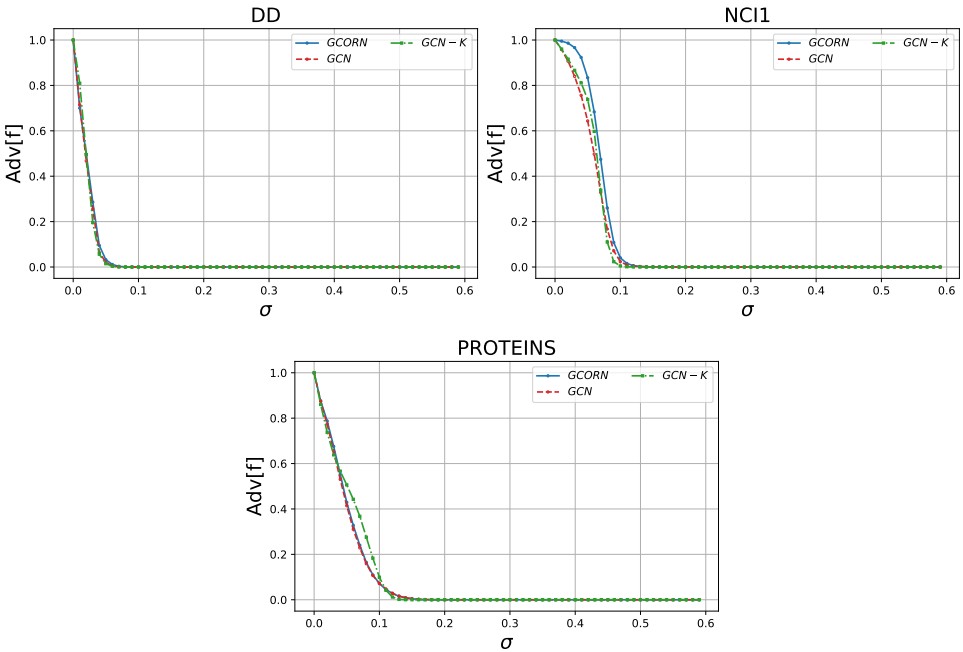

Figure 7: The Values of $Adv_\epsilon^{\alpha,\beta}[f]$ for the dataset CiteSeer, PubMed and CS.

