# OpenReview forum: "Bounding the Expected Robustness of Graph Neural Networks Subject to Node Feature Attacks"
_ICLR.cc/2024/Conference — ICLR 2024 poster_

### Official Review · Reviewer_nkA1 · 2023-10-31

**Soundness:** 2 fair
**Presentation:** 3 good
**Contribution:** 2 fair
**Rating:** 5
**Confidence:** 3

**Summary:**

This paper studies the robustness of GNNs under adversarial attacks to node features in attributed graphs. The authors establish the upper bound for the expected robustness of GCNs under the node feature perturbations, and generalize to GIN as well as structural attacks. The bound also motivates a solution called Graph Convolutional Orthogonal Robust Networks (GCORNS), that adopts  orthogonal projections of the weight matrices of GNNs to reduce the upper bound (and to improve the robustness). Moreover, the authors also propose an estimation method of the proposed robustness measure. Then they conduct experiments to demonstrate the improved robustness of GCORN method.

**Strengths:**

- The proposed robustness measure is new;

- The proposed method is somehow interesting;

- The presentation and organization are clear;

**Weaknesses:**

(-) The scope seems to be limited;

(-) Some baselines have not been compared with.

(-) The improvements seem to be incremental and limited;

**Questions:**

1. The scope seems to be too limited:
a. If the scope is as claimed as focusing on adversarial attacks on node features, what are the practical scenarios? Are there any realistic cases for the studied attack?

b. As already been shown by the authors, there is a natural connection between structural attacks and the attribute attacks, why not study the more general attacks?

c. The work also neglects a rich literature in the line of graph injection attack, where the adversary will inject new nodes and optimize the injected nodes’ features to perform the attack [1,2,3,4]. It is also worth discussing the connections of the proposed robustness measure with respect to injection attacks, as well as the recently emerging backdoor attacks [5,6].

2. There are already multiple new robustness measures proposed [3,7] which have not been discussed in the paper.

3. Some baselines have not been compared with, including [8,9,10]. Can GCORNS outperform them?

b. Besides, why table 1 and table 2 adopt different baselines?

c. Can a heuristic solution like weight decay achieve the same functionality as GCORNS?

d. Can GCORNS be incorporated to more GNN backbones other than GCN?

e. Would GCORNS work for large graphs?

4. The improvements seem to be limited when with slightly more powerful attacks. For example, in Table 1, with Nettack, GCORN underperform AirGNN in ogbg-arxiv, and RGCN in PubMed. In Table 2, with structural attacks, GCORN underperform previous methods in even more cases.


**References**

[1] TDGIA: Effective Injection Attacks on Graph Neural Networks, KDD’21.

[2] Single Node Injection Attack against Graph Neural Networks, CIKM’21.

[3] Understanding and Improving Graph Injection Attack by Promoting Unnoticeability, ICLR’22.

[4] Let Graph be the Go Board: Gradient-free Node Injection Attack for Graph Neural Networks via Reinforcement Learning, AAAI’23.

[5] Unnoticeable Backdoor Attacks on Graph Neural Networks, WWW’23.

[6] On Strengthening and Defending Graph Reconstruction Attack with Markov Chain Approximation, ICML’23.

[7] Revisiting Robustness in Graph Machine Learning, NeurIPS’22.

[8] Graph Structure Learning for Robust Graph Neural Networks, KDD’20.

[9] Elastic graph neural networks, ICML’21.

[10] EvenNet: Ignoring Odd-Hop Neighbors Improves Robustness of Graph Neural Networks, NeurIPS’22.

---

> ### Author Response · Authors · 2023-11-21
> **Response to nkA1's review (1/3)**
>
> We thank the reviewer for his comments and question that helped improve the quality of our manuscript. In what follows, we address the raised questions and weaknesses point-by-point.
>
> **[Q1] On the limitation of the considered scope and application of node features attacks:**
> We disagree with the reviewer on the fact that focussing on node features attacks is too limiting. We start by noting that while adversarial attacks may be deliberately crafted by a user, understanding and defending them also sheds light on how to interact with abnormal node features which are frequent in real-world datasets. Regarding the possible applications, we can consider social networks, in which a user can choose to edit/add their profile information (which for the more widely used social networks is easier than adding/deleting connections corresponding to the structural perturbation case) in order to get recommended to all the users (which is an edge-prediction task) or escape scam detection (which is a node classification task). Similarly, a company can enter/fake its product information in a marketplace such as to get shown to all the potential users using the underlying recommender system (which is an edge prediction task) or be listed in the wrong category (which is a node classification task). A more global and important perspective lies within the cybersecurity industry, where we often consider a network in which each node represents a computer/user with the edges representing the probability of observing the connection between the nodes. In order to enhance the modeling, additional node features are added such as the number of bytes exchanged, user credentials, number of connections with other users, which are externally controlled parameters. The goal in cyber-security is to estimate how rare these connections are, consequently ensuring the viability of the connection. In this context data poisoning attacks are common. In these attacks a user simply edits node features (where for instance a user will deliberately edit certain aspects of their credentials) to fool our security firewall/classifier.
>
> **[Q2] Regarding the general case of attacking the structure and node features:**
> We apologize if we didn’t specify this enough in the original manuscript but we have actually provided a theoretical analysis on the general attack where the structure and node features are both attacked in parallel (Appendix F). We note that we have furthermore added some experiments (Table 3 - Appendix F) where our models outperforms the other benchmarks when dealing with both the attacks (Mettack & Random) in parallel. We thank the reviewer for pointing this out.
>
> Table 1: Performance of GCORN and defense benchmarks when subject to both structural and node features adversarial attacks in parallel.
> | Dataset  | GCN              | GCN-Jaccard    | RGCN           | GNN-SVD        | GNNGuard       | ParsevalR      | GCORN                   |
> |----------|------------------|----------------|----------------|----------------|----------------|----------------|-------------------------|
> | Cora     | 76.7   $\pm$ 1.2 | 76.7 $\pm$ 0.4 | 78.1 $\pm$ 0.6 | 65.9 $\pm$ 4.0 | 61.0 $\pm$ 0.4 | 77.6 $\pm$ 1.1 | **79.8 $\pm$ 0.7** |
> | CiteSeer | 68.7 $\pm$ 0.3   | 71.5 $\pm$ 0.3 | 68.9 $\pm$ 0.8 | 68.3 $\pm$ 0.5 | 65.0 $\pm$ 0.3 | 70.9 $\pm$ 0.6 | **74.2 $\pm$ 0.3** |
> | PubMed   | 47.0 $\pm$ 0.4   | 49.4 $\pm$ 0.8 | 51.2 $\pm$ 0.4 | 47.7 $\pm$ 0.5 | 40.3 $\pm$ 0.3 | 49.7 $\pm$ 1.1 | **53.9 $\pm$ 0.8** |
> | CoraML   | 63.3 $\pm$ 0.0   | 32.2 $\pm$ 1.5 | 59.7 $\pm$ 1.4 | 53.8 $\pm$ 0.8 | 32.2 $\pm$ 1.5 | 65.4 $\pm$ 0.2 | **77.4 $\pm$ 1.1** |
>
> **[Q3] Regarding graph injection attack techniques:**
> We appreciate the reviewer's valuable input, and we have incorporated the references and the discussion into the related work section of the revised manuscript. The concept of "node injection attacks" indeed presents an intriguing perspective on adversarial techniques. Our current study predominantly explores "classical" attacks and our theoretical analysis is dependent on the introduced graph distance. In the context of node injection attacks, a new graph distance should be introduced/adapted to measure the similarity between graphs (with different number of nodes - giving the injection) and the concept of input’s neighborhood should also be revisited. This adaptation indeed is an exciting avenue for extending our work.

---

> ### Author Response · Authors · 2023-11-21
> **Response to nkA1's review (2/3)**
>
> **[Q4] Regarding other robustness measure [3, 7]:**
> We have added these references in our related work section. These two works are more interested in the structural aspect of the attacks where they consider the possible semantic content change in the adversarial graphs. While we agree that considering these evaluations is important, we also note that in addition to our proposed robustness evaluation we have considered the robustness evaluation provided by [1] (Figure 1.c) which allows us to consider both structural and node features based on robustness estimation.
>
> [1] A. Bojchevski et Al. - Efficient Robustness Certificates for Discrete Data: Sparsity-Aware Randomized Smoothing for Graphs, Images and More, ICML 2020.
>
> **[Q5] Comparing GCORN to the provided baselines [8, 9, 10]:**
> We thank the reviewer for pointing out these references, we have added experiments related to several of these models in the updated manuscript (Table 6 - Section J in the Appendix) and we list them also in this response. We note that the reference EvenNet work is very interesting and very relevant to our perspective, where it shows that actually ignoring some “odd-hop neighbors” improves the robustness. This is directly related to the second part of our computed bound (related to the normalized walks over the input graph). So while we have focused on the first part of the bound (related to the norm weights), this work seems to directly prove our second part of the bound. We finally would like to point out that our method is attack-agnostic since our theoretical design and analysis is considering the input graph’s whole neighborhood.
>
> Table 2: Performance of GCORN, ElasticGNN and EvenNet when subject to node-features adversarial Attacks.
> | Attack | Dataset  | ElasticGNN     | EvenNet        | GCORN          |
> |--------|----------|----------------|----------------|----------------|
> | Random ($\psi =0.5$) | Cora     | 72.5 $\pm$ 2.4 | 69.7 $\pm$ 1.8 | 77.1 $\pm$ 1.8 |
> |        | CiteSeer | 58.9 $\pm$ 2.3 | 48.6 $\pm$ 1.6 | 67.8 $\pm$ 1.4 |
> |        | PubMed   | 70.5 $\pm$ 0.6 | 68.8 $\pm$ 1.6 | 73.1 $\pm$ 1.1 |
> | Random ($\psi =1.0$) | Cora     | 54.4 $\pm$ 2.4 | 48.9 $\pm$ 2.1 | 57.6 $\pm$ 1.9 |
> |        | CiteSeer | 43.8 $\pm$ 2.0 | 39.7 $\pm$ 4.2 | 57.3 $\pm$ 1.7 |
> |        | PubMed   | 56.8 $\pm$ 2.1 | 58.5 $\pm$ 3.2 | 65.8 $\pm$ 1.4 |
> | PGD    | Cora     | 64.7 $\pm$ 1.6 | 57.6 $\pm$ 3.6 | 71.1 $\pm$ 1.4 |
> |        | CiteSeer | 50.8 $\pm$ 3.2 | 37.1 $\pm$ 3.3 | 65.6 $\pm$ 1.4 |
> |        | PubMed   | 64.6 $\pm$ 1.5 | 61.0 $\pm$ 4.1 | 72.3 $\pm$ 1.3 |
>
> **[Q6] On the difference between adopted baselines in Table 1 & 2:**
> We apologize for any potential confusion. Table 1 focuses on node feature-based adversarial attacks (to validate the results of Theorem 4.1), whereas Table 2 addresses structural perturbations (related to Theorem 4.2). Consequently, our choice was to align each defense benchmark with the corresponding specific attack that it aims to defend. Using defense benchmarks with attack types beyond their original score resulted in poor performance and is therefore a penalization to these methods compromising the validity of our evaluation.
>
> **[Q7] On possible use of heuristic solution like weight decay:**
> Based on the computed upper-bound (Theorem 4.1 & 4.2), weight decay which consists of adding a penalization term to regularize the weights can indeed be a solution to tighten the upper-bound resulting in an enhanced robustness. This is similar to what has been proposed by the benchmark Parseval Network (ParsevalR). While we have not explicitly considered weight decay, from the experimental results on ParsevalR and the use of some regularization techniques, we have seen that the related constraints to these methods usually deeply affects the model’s clean accuracy. And since apriori, we don’t know if the input graph is attacked or not, seeking a good balance between robustness and clean accuracy is crucial, hence the choice of our projection technique.

---

> ### Author Response · Authors · 2023-11-21
> **Response to nkA1's review (3/3)**
>
> **[Q8] Incorporating GCORN to other GNN backbones other than GCN:**
> Although our primary focus is centered on the GCN model, we know that our approach can be extended to various GNN architectures and we have indeed outlined theoretical insights and guidelines in Appendix E aiming to generalize our theoretical investigation to diverse message-passing models. Moreover, in the original manuscript, we established the theoretical upper-bound for GIN (Theorem 4.3) and we additionally supplement this theoretical analysis with an experimental analysis that shows similar enhancement for the GIN as the one witness for the GCN model as shown by the following table. Note that we added this table to the revised manuscript (Table 8 - Appendix K).
> Table 3: Performance of our proposed method GIORN and GIN when using a GIN backbone  when subject to both features adversarial attacks.
>
> |                      | Model | Cora                    | CiteSeer                | PubMed                  |
> |----------------------|-------|-------------------------|-------------------------|-------------------------|
> | Random ($\psi =0.5$) | GIN   | 67.8 $\pm$ 1.3          | 50.1 $\pm$ 0.3          | 72.8 $\pm$ 0.4          |
> |                      | GIORN | **71.8 $\pm$ 0.8**   | **60.8 $\pm$ 0.5** | **74.3 $\pm$ 0.7** |
> | Random ($\psi =1.0$) | GIN   | 54.5 $\pm$ 0.9          | 45.1 $\pm$ 0.4          | 67.3 $\pm$ 0.6          |
> |                      | GIORN | **66.7 $\pm$ 1.2** | **54.4 $\pm$ 0.8** | **68.9 $\pm$ 1.1** |
> | PGD                  | GIN   | 61.4 $\pm$ 1.4          | 44.1 $\pm$ 0.5          | 70.6 $\pm$ 0.4          |
> |                      | GIORN | **69.5 $\pm$ 0.6** | **58.3 $\pm$ 0.6** | **73.1 $\pm$ 1.6** |
>
> **[Q9] Would GCORN work for large graphs?**
> This is indeed an important point which we tried to address in Section 4.3 (“Complexity of our method”). We note that the complexity introduced by the modification we make in our method is independent of the input graph’s size since our method is mainly concerned with the weight projection into the orthogonal manifold, depending therefore on the size of the hidden dimension (which is very small for the majority of dataset - 16 for Cora for instance). Hence, the complexity of the GCORN and GCN grow equally as the graph size grows. This is not the case of the majority of available defense benchmarks where for instance the closely performant method to ours in structural perturbations which is GNNGuard has a complexity increasing with the input graph. Consequently, our method GCORN scales more gracefully to large graphs than  available techniques. We note that we additionally provided complimentary time/complexity analysis in Appendix G.
>
> **[Q10] On the limited performance improvement over other baselines in certain cases.**
> Firstly, we want to highlight that the primary strength of our paper lies in its theoretical analysis and guarantees, aspects notably absent in most of the benchmarks considered. These facets underscore the potential effectiveness of our approach against future, more sophisticated attacks. Our experimental evaluation shows that, in the majority of the cases,  our method outperforms all benchmarks (which are considered state-of-the art defenses) across various scenarios—node features, structural, and mixed attacks. In the cases where one of the several high-performing benchmarks displays better results than our GCORN method, the two methods are often within the range of standard deviation of each other. Regarding Nettack in particular, it is crucial to note that Nettack is a targeted attack. Consequently, the evaluation focused on a limited number of nodes (40 in this instance, as per the original work). Therefore, we expect the observed enhancement gap not be substantial, which differs from other considered attacks that approach the attack aim from a global perspective attacking all the nodes (and in which the observed performance gap is important). Finally, using our robustness estimation (based on our robustness formulation) in Figure 1 (a) & (b), we can see the real robustness gap between our proposed GCORN and the other benchmarks, which is rather related to the method itself and not the considered attack (since we evaluate the whole input graph’s neighborhood).

---

### Official Review · Reviewer_B7fh · 2023-11-08

**Soundness:** 4 excellent
**Presentation:** 3 good
**Contribution:** 3 good
**Rating:** 8
**Confidence:** 2

**Summary:**

The paper proposes a new formalization of the notion of expected robustness for GNN and introduce some theoretical guarantees on this quantity. It also uses the insights coming from the bound in the case of GCN to propose a robust alterntive under the form of GCORN. The paper also validates the relevance of their approach empirically.

**Strengths:**

* The paper is well structured and easy to read
* The paper proposes a clear and general framework for adversarial robustness on graphs
* The paper derives some theoretical bounds for expected adversarial robustness and leverage this formulation to propose some simple but efficient improvement based on orthonormalisation with the GCORN model
* In the case of MPNN, the paper provides both bounds for node features attacks and structural attack
* It also validates that his approach work in practice

**Weaknesses:**

* The paper considers only a few attacks to validate his approach. Even if these attacks makes sense (PGD / Nettack are good standards for this type of tests), it would have been interesting to try additional ones.
* The gain of robustness of GCORN over GCN is interesting but it's not clear how it compares to other defense methods (for instance is GCORN w adversarial training still better than GCN with adversarial training)
* It is unclear to me how tight is the bound from theorem 4.1/4.2 ; this part could be stated more clearly in the experiment section.

**Questions:**

* Do you have some table summarizing the tightness of the bound in the experiments you ran ?
* You focus mainly on the cases where alpha, beta=(0,1) or (1,0). Do you have some insights on the more general case of mixed attacks (structural and feature based) ? Are some results holding in that case ? (combination of 4.1 and 4.2 ?)

---

> ### Author Response · Authors · 2023-11-21
> **Response to B7fh's review**
>
> We thank the reviewer for their strong support recognizing the novelty and application of our work. We additionally are grateful for their comments and questions that we aim to address point-by-point in what follows.
>
> **[Q1 & W3] On the tightness of the provided and computed upper-bound:**
> We agree that this was missing from our original manuscript. The focus of our work is to derive an upper-bound, which when tightened allows us to propose a model with enhanced inherent robustness. Regarding the tightness of the bound, we have experimentally studied this quantity by computing the theoretical upper-bound together with our estimation of the expected adversarial robustness for the Cora dataset. We provided the results in the updated manuscript in Figure 5 in Appendix M.3. While we observe the bound to not be tight, we see that it is relatively close to the empirically estimated robustness. The fact that the theoretical bound and the estimated expected robustness are meaningfully close, is further substantiated by the fact that reducing the upper-bound allowed us to improve model robustness in the experiments shown in Tables 1 and 2.
>
> **[Q2 & W1] Insights on the more general case of mixed attacks (combining Structural and node feature attacks):**
> In fact in Appendix F of our original submission we provided a theoretical analysis (in form of a bound on the expected robustness) for the case in which both the structure and the node features are attacked concurrently ($\beta = (1,1)$). We additionally added experiments to compare the behavior of our proposed GCORN and other defense benchmarks when dealing with the mixed attacks (Mettack & Random) in parallel illustrating our theoretical analysis. Table 1 is providing these results and they have been also added to the revised manuscript (Table 3 - Appendix F). We thank the reviewer for pointing this out.
>
> Table 1: Performance of GCORN and defense benchmarks when subject to both structural and node features adversarial attacks in parallel.
> | Dataset  | GCN              | GCN-Jaccard    | RGCN           | GNN-SVD        | GNNGuard       | ParsevalR      | GCORN                   |
> |----------|------------------|----------------|----------------|----------------|----------------|----------------|-------------------------|
> | Cora     | 76.7   $\pm$ 1.2 | 76.7 $\pm$ 0.4 | 78.1 $\pm$ 0.6 | 65.9 $\pm$ 4.0 | 61.0 $\pm$ 0.4 | 77.6 $\pm$ 1.1 | **79.8 $\pm$ 0.7** |
> | CiteSeer | 68.7 $\pm$ 0.3   | 71.5 $\pm$ 0.3 | 68.9 $\pm$ 0.8 | 68.3 $\pm$ 0.5 | 65.0 $\pm$ 0.3 | 70.9 $\pm$ 0.6 | **74.2 $\pm$ 0.3** |
> | PubMed   | 47.0 $\pm$ 0.4   | 49.4 $\pm$ 0.8 | 51.2 $\pm$ 0.4 | 47.7 $\pm$ 0.5 | 40.3 $\pm$ 0.3 | 49.7 $\pm$ 1.1 | **53.9 $\pm$ 0.8** |
> | CoraML   | 63.3 $\pm$ 0.0   | 32.2 $\pm$ 1.5 | 59.7 $\pm$ 1.4 | 53.8 $\pm$ 0.8 | 32.2 $\pm$ 1.5 | 65.4 $\pm$ 0.2 | **77.4 $\pm$ 1.1** |

---

### Official Review · Reviewer_dcsE · 2023-11-10

**Soundness:** 3 good
**Presentation:** 4 excellent
**Contribution:** 3 good
**Rating:** 6
**Confidence:** 4

**Summary:**

In this paper ,the authors theoretically define the concept of expected robustness to evaluate the robustness of GNNs, extending beyond the traditional "worst-case" adversarial robustness, and derive an upper bound of the expected robustness of GCNs and GINs. They propose a modification to the GCN architecture called the Graph Convolutional Orthogonal Robust Network (GCORN) that aims to improve robustness against node feature perturbations by promoting the orthogonality of weight matrices, effectively controlling the norm of these matrices to mitigate the impact of such attacks. The authors employ an iterative orthogonalization process that also benefits learning convergence by mitigating vanishing and exploding gradients. They ensure that the complexity of their approach remains manageable and does not excessively increase with larger graph sizes. Furthermore, they develop a probabilistic and model-agnostic method for empirical evaluation of GNN robustness. Unlike existing metrics, this proposed method applies to a variety of attack types and does not require the knowledge of specific attack schemes. They present experimental results to demonstrate the superior robustness of GCORN compared to baseline models, using various real-world datasets. The findings suggest that GCORN is a promising modification to enhance GNNs' robustness to adversarial feature perturbations without significant trade-offs in performance.

**Strengths:**

Originality: The paper introduces a new theoretical framework for defining and quantifying the concept of "Expected Adversarial Robustness" for GNNs, which is a deviation from the traditional worst-case scenario evaluation that dominates the field. Their GCORN model demonstrates an innovative approach to mitigating the impact of adversarial attacks on GNNs by leveraging the orthogonality of weight matrices, a concept that has not been extensively explored in the context of GNNs. The development of a model-agnostic, probabilistic method for evaluating GNN robustness is a creative combination of existing ideas that significantly enhances the security assessment of graph models against a wider range of adversarial strategies.
Quality: The quality of the paper is high as it thoroughly examines the proposed concepts from both theoretical and empirical perspectives. It provides a rigorous mathematical formulation of the problem and strong theoretical foundations for the methods proposed.
Clarity: The paper is well-written and organized in a manner that strategically guides the reader through both the theoretical and practical aspects of the work. Definitions and theoretical findings are clearly presented and sufficiently detailed, making them accessible to readers with a foundational understanding of the domain.
Significance: The Expected Adversarial Robustness framework gives practitioners a more nuanced understanding of model vulnerability in realistic scenarios beyond the worst-case adversarial examples.
In summary, the paper showcases original conceptual developments, high-quality theoretical work, clarity in its exposition, and significant contributions to the field of graph representation learning, particularly addressing adversarial robustness in GNNs.

**Weaknesses:**

While the paper presents significant contributions to the stability and robustness of GNNs against feature-based adversarial attacks, there are certain areas where it could potentially be improved to reinforce its claims and widen its applicability:
1. Computational Cost: The estimation of a GNN’s expected robustness involves a sampling-based approach, which indeed can be computationally intensive as it requires generating and evaluating numerous perturbed versions of the input graph, especially when it comes to large datasets.
2. The robustness measure might vary significantly with each estimation due to sampling, leading to inconsistency and making it difficult to compare with other models.
3. The paper lacks the comparison between the theoretical upper bound of the expected robustness and its empirical estimates. The authors have provided a theoretical upper bound for the expected adversarial robustness of GNNs and introduced an empirical method to estimate this robustness; however, the paper does not explicitly show how closely the empirical results align with the theoretical bound.
4. Attack Models and Benchmarks: It appears that the attack models used to evaluate robustness are largely selected from existing and potentially well-known strategies. Exploration of the effectiveness of GCORN against emerging or more sophisticated adversarial models could further substantiate the claimed robustness. Furthermore, including newer or well developed defense mechanisms would be advantageous for our understanding of GCORN’s effectiveness.

**Questions:**

1. How to determine the number of samples required to obtain an accurate empirical estimation of the robustness? Is there a theoretical guarantee (such as a bound on the estimation error with high probability) that the estimated value is close enough to the true expected robustness?
2. A comparison between the theoretical upper bound of the expected robustness and its empirical estimates should be provided in order to show how closely the empirical results align with the theoretical bound and the effectiveness of the theoretical bound.
3. It is encouraged to include experiments with more sophisticated adversarial models and comparisons with other defense methods.

---

> ### Author Response · Authors · 2023-11-21
> **Response to dcsE's review (1/2)**
>
> We thank the reviewer for their thoughtful comments, which have allowed us to improve the quality of our manuscript and to post a revised version. We are glad they identified the originality and applicability of our proposed approach. In what follows, we address the raised questions and weaknesses point-by-point.
>
> **[W1] Regarding the computational cost:** As discussed in Section 5, Appendix H and I, the computational cost of estimating a GNN’s expected robustness equals the time of firstly, sampling perturbed attributed graphs and secondly, running inference of the studied GNN on the perturbed graphs (potentially in parallel or in batches). Firstly, we sample perturbed attributed graphs by sampling from uniform distributions, which has a complexity of O(1). Secondly, the complexity of performing inference of a GNN on several graphs depends on the chosen GNN, for the GCN for example the inference complexity is linear in the number of edges in the perturbed graphs.
>
> With regards to step 1 of our evaluation procedure, existing evaluation approaches often evaluate model accuracy by injecting Gaussian noise [5], the generation of which is of, e.g., $O((nK)^3)$ for $n$ $K$-dimensional samples drawn when using the method relying on the Cholesky decomposition [1]. Now concerning step 2 of our evaluation procedure, using the GNN for inference several times is required for the large majority of traditional robustness evaluations, e.g. in the Project Gradient Descent (PGD) algorithm [2], we have to perform an inference at each iteration of the algorithm. Furthermore, each iteration involves computing the gradient and updating the input. The number of iterations needed for convergence in the PGD method can be high, leading to increased computational cost [3,4]. Consequently, our proposed expected robustness metric has a lower computational cost than the PGD metric.
>
> We therefore reckon the computational cost of our estimation procedure to be limited and certainly lower than most existing robustness evaluation approaches.
>
> [1] T. Muschinski, G. Mayr, T. Simon,  N. Umlauf & A. Zeileis - Cholesky-based multivariate Gaussian regression, Econometrics and Statistics 2022.
>
> [2] K. Xu, H. Chen & Al. - Topology attack and defense for graph neural networks: An optimization perspective, IJCAI 2019.
>
> [3] J. Tian, Y. Liu, J. Smith, Z. Kira - Fast Trainable Projection for Robust Fine-Tuning, NeurIPS 2023
>
> [4] J. Tian, X. Dai, C. Ma, Z. He, Y. Liu Z. Kira - Trainable Projected Gradient Method for Robust Fine-tuning, CVPR 2023.
>
> [5] B. Li, C. Chen, W. Wang, L. Carin - Certified Adversarial Robustness with Additive Noise - NeurIPS 2019
>
> **[Q1 & W2] On the effect of number of samples on the robustness estimation:**
> We agree with the reviewer that analyzing the effect of the number of samples on the variance of our robustness estimation is interesting. In response, we have conducted both theoretical and empirical work to answer your question.
>
> Firstly, we theoretically studied the minimum number of samples that could allow us to converge to a reliable estimate. We hence considered the concept of discrepancy of volumes and its known link with the quality of estimation [1, 2]. We provided a complete analysis in the updated manuscript in Appendix M.2 where we derive the following inequality:
>
> $ \hspace{8cm} L_{max}  \geq \frac{log(\alpha) }{log\left ( 1  -  \left ( \frac{ r}{ \epsilon} \right )^K \right )}.$
>
> Recall, that $L_{max}$ denotes the number of sampling per graph, $r$ the radius, $K$ the feature dimensions, $\epsilon$ the attack budget (also defining the input graph’s neighborhood), and $\alpha$ is a discrepancy-confidence level.
> This bound demonstrates that there exists a required minimum number of samples in order to reach a low discrepancy of volumes, which corresponds to a lower estimation error. The functional form of the bound confirms the intuition that with a small $\epsilon$, we need fewer samples to estimate the robustness of the underlying model.
>
> We furthermore empirically studied the number of samples required for an acceptably low variance of the robustness estimate. Figure 4 in Appendix M.2 displays the results of this study for the Cora dataset. We find that the estimate is already reasonably accurate for small sample sizes, even for 10 samples only, and for the standard deviation to meaningfully decrease within the first 40 considered samples. Thus, with only a small number of samples, we can accurately estimate the expected robustness in practice.
>
> [1] H. Niederreiter - Random number generation and quasi-Monte Carlo methods, SIAM 1992.
>
> [2] S. Tezuka - Uniform random numbers: Theory and practice, Springer Science & Business Media 2012.

---

> > ### Author Response · Authors · 2023-11-21
> > **Response to dcsE's review (2/2)**
> >
> > **[Q2 & W3] On the theoretical and computed bound:**
> >
> > Firstly, the focus of our work is to derive an upper-bound, which when tightened allows us to propose a model with enhanced inherent robustness. Yet, as requested we have conducted an experimental assessment where, for a specific model, we calculated both its theoretical upper-bound and the estimated robustness evaluation (using sufficient number of samples as explained in our answer to Q1). The results are provided in the updated manuscript in Figure 5 in Appendix M.3. While we observe the bound to not be tight, we see that it is relatively close to the empirically estimated robustness. The fact that the theoretical bound and the estimated expected robustness are meaningfully close, is further substantiated by the fact that reducing the upper-bound allowed us to improve model robustness in the experiments shown in Tables 1 and 2.
> >
> > **[Q3 & W4] On the considered attack/defense strategies and benchmarks:**
> > We thank the reviewer for this suggestion. We have added further benchmarks that were proposed by Reviewer 4  in Table 6 - Section J in the Appendix. We furthermore agree with the reviewer on the importance of developing more advanced attacks, which are, to the best of our knowledge, not available in the literature for now. We also want to remark that one of the main advantages of our method is our theoretical choice to focus on the complete input graph’s neighborhood rather than specific worst-case examples (which is by default a consequence - Lemma 3.2). Consequently, our expected robustness improvement and evaluation is attack-agnostic. The most-commonly followed robustness evaluation in the literature is based on using the benchmark, i.e., most performant, adversarial attacks and while some defense methods can successfully defend against them, no guarantee on their performance is ensured for future more advanced attacks which is not the case of our approach (since by default new more advanced attacks resides within the considered neighborhood in which we have a guaranteed robustness upper-bound). We finally note that in our experimental analysis and in addition to our proposed evaluation (which we consider an alternative to the previously discussed point), we have used the benchmark attacks that are usually used in the literature (PGD, Nettack, Random, Mettack, DICE). If the reviewer is aware of specific baselines, that they think would improve our empirical evaluation, we would be glad to include them.

---

### Official Review · Reviewer_gkkc · 2023-11-11

**Soundness:** 4 excellent
**Presentation:** 3 good
**Contribution:** 3 good
**Rating:** 8
**Confidence:** 3

**Summary:**

In this paper, authors have provided theoretical as well as empirical study of vulnerability of GNNs to adversarial attacks. The concept of "Expected Adversarial Robustness" for GNNs is introduced and studied in relation to conventional adversarial robustness. An upper bound of this robustness concept is derived and proved for the networks which is independent of the model or attack. Based on these results, GCORN is introduced which is a training method to improve robustness of graph networks by controlling the norm of weight matrices, encouraging their orthonormality. Empirical analysis is conducted to illustrate the effectiveness of GCORN in comparison to other defense baselines.

**Strengths:**

**Originality**: This work introduces and studies the concept of expected robustness of graph networks and sounds very novel. Not only strong theoretical foundation is provided, methods are proposed to empirically calculate the robustness. Furthermore based on findings, a novel method to improve robustness is introduced. The theoretical and empirical claims speak to the novelty of the work.

**Quality**: The paper is characterized by its rigorous theoretical exploration and empirical analysis, which collectively elevate its overall quality. The experimental results strongly support the theoretical claims.

**Clarity**: The concepts in the paper are clear and well organized. The narrative flow keeps the reader engaging and this clarity in presentation makes the complex subject matter accessible and comprehensible.

**Significance**: I think this work provides significant insights to the robustness of GNNs and will be useful for the research community. The GCORN method is effective, has good theoretical foundation and adds a new benchmark for training robust GNNs.

**Weaknesses:**

A few typos:
Section 4.3 encourages -> encourage
Algorithm 1: Second for loop line# 3 $X +Z$ should be $X_i + Z_i$?

**Questions:**

see weaknesses

---

> ### Author Response · Authors · 2023-11-21
> **Response to gkkc's review**
>
> We would like to thank the reviewer for their positive and encouraging review. We are pleased that they liked the novelty and applicability of our proposed approach. We additionally are grateful to the reviewer for spotting these typos. Indeed, in the inner loop of Algorithm 1, the features vector $X$ should be replaced by $X_i$ as we may have different graphs in the dataset, e.g. in the graph classification task. The quantity $Z$ is resampled at each iteration of the inner loop in Step 2 and hence, the algorithm is correctly stated without subscripting $Z.$ However, to avoid any confusion, we have chosen to replace $Z$ by $Z_l,$ where $l$ is the iteration counter of the inner loop,  in Steps 2 and 3 of Algorithm 1. We have made the necessary adaptations in our revised manuscript.

---

### Meta-Review · Area_Chair_eymR · 2023-12-11

**Metareview:**

This paper proposes a novel approach to measuring the expected robustness of graph neural networks (GNNs). By leveraging a derived robustness upper bound, the authors motivate a defense algorithm that enforces the weights to be orthonormal during training. While the individual components of this work, such as the bounds presented in Section 4.1 and the concept of restricting weight eigenvalues, have been explored for neural networks in the past, this paper makes a significant contribution by formally demonstrating their applicability to GNNs. Notably, the resulting robustness estimation procedure and defense algorithm exhibit a computational complexity dependent solely on the hidden dimension instead of the graph size, enabling the proposed method to effectively handle large graphs. Most reviewers commend the paper's solidity and recommend acceptance. Reviewer nkA1's technical questions have also been addressed to a satisfactory degree. Therefore, we recommend accepting this paper.

**Justification For Why Not Higher Score:**

Several ideas in this paper has been studied for other types of neural networks, so the contributions may not be sufficient for  splotlight/oral.

**Justification For Why Not Lower Score:**

This paper presents a solid contribution to adversarial defense for graph neural networks.

---

### Decision · Program_Chairs · 2024-01-16

Accept (poster)